

# Full counting statistics in the transverse field Ising chain

**Stefan Groha[1]⋆, Fabian Essler[1] and Pasquale Calabrese[2,3]**

**1** The Rudolf Peierls Centre for Theoretical Physics, Oxford University, Oxford, OX1 3NP, UK
**2** SISSA and INFN, via Bonomea 265, 34136 Trieste, Italy
**3** International Centre for Theoretical Physics (ICTP), I-34151, Trieste, Italy

⋆ stefan.groha@physics.ox.ac.uk

## Abstract

We consider the full probability distribution for the transverse magnetization of a finite subsystem in the transverse field Ising chain. We derive a determinant representation of the corresponding characteristic function for general Gaussian states. We consider applications to the full counting statistics in the ground state, finite temperature equilibrium states, non-equilibrium steady states and time evolution after global quantum quenches. We derive an analytical expression for the time and subsystem size dependence of the characteristic function at sufficiently late times after a quantum quench. This expression features an interesting multiple light-cone structure.


---

# 1 Introduction

The statistical nature of measurements of observables is a fundamental principle of quantum mechanics. Measuring the same observable in identically prepared systems leads to different measurement outcomes that are described by a probability distribution that depends on both the state $|\Psi\rangle$ and on the observable $\mathcal{O}$ considered. The full probability distribution $P(\mathcal{O}, |\Psi\rangle)$ encodes detailed information about quantum fluctuations in the system. It is of particular interest in situations where the first few moments do not provide a good description of the distribution. Quantum mechanical probability distributions in the guise of *Full Counting Statistics* (FCS) have been studied for some time in mesoscopic devices [1,2]. More recently it has become possible to analyze them in systems of ultra-cold atomic gases [3–8]. This has broken new ground in the sense that one is dealing with (strongly) interacting many-particle systems and a variety of observables, typically defined on subsystems, can be accessed. This has motivated a number of theoretical works of FCS in equilibrium states [9–20], and after quantum quenches [7, 20–24]. A second motivation for studying FCS has been the observation that in non-interacting fermionic systems with particle number conservation the FCS of particle number within a subsystem is directly related to the entanglement entropy [25–34] and provides indirect information about the latter.

From a theoretical point of view calculating the FCS for a given observable on a sizeable subsystem poses a formidable problem and as a result only very few exact results are available even in simple equilibrium situations. Even less is known about FCS after quantum quenches. This motivates reconsidering FCS in the transverse field Ising chain (TFIC). The TFIC is a key paradigm for quantum phase transitions [35] and a simple, but non-trivial, many-body system without particle number conservation and therefore provides an ideal playground for studying

FCS both in and out of equilibrium. Indeed, thanks to the mapping of the TFIC to a model of non-interacting spinless fermions with pairing term it is possible to analytically determine ground state and thermal properties, see e.g. [35–37], as well as describe the non-equilibrium dynamics of local observables [38–44] and of the reduced density matrix of a block of adjacent sites [43, 45–47] after a global quantum quench. A summary of these developments is given in the recent reviews Refs [48, 49].

In this work we focus on the FCS of the simplest observable, the transverse magnetization within a block of $\ell$ adjacent spins. In the ground state this problem has been previously analyzed in Refs [9, 11] and generic Gaussian states have been considered as well [14]. We note that the ground state FCS of the longitudinal magnetization at the critical point has been determined in Ref. [10] and the ground state FCS of the subsystem energy was considered in Ref. [18].

This manuscript is organised as follows. In section 2 we first introduce the TFIC and briefly summarize the important steps for diagonalizing the Hamiltonian. We then define the FCS and the associated generating functions considered in this work. In section 3 we provide a novel derivation of an efficient determinant representation for the FCS in general $\mathbb{Z}_2$ invariant Gaussian states. The result is equivalent to that of Ref. [14]. This result is applied in section 4 to the determination of the FCS in equilibrium states. In the ground state we recover the results of Ref. [9]. Our results for the FCS in finite temperature equilibrium states are to the best of our knowledge new. In section 5 we turn to the main point of interest: the time evolution of the FCS after a global quantum quench. We consider the situation where the system is prepared in a pure state at a finite finite energy density and then time evolved with a Hamiltonian $H$ that does not commute with the initial state density matrix, which leads to non-trivial dynamics. We present explicit results for general "transverse field" quenches as well as evolution starting in a classical Néel state. The main result of this work is presented in section 6: an analytic expression for the time evolution of the FCS after a transverse field quench. In section 8 we summarize our results and comment on a number of issues that deserve further investigation.

## 2 The model and the full counting statistics

### 2.1 Transverse Field Ising chain

In the following we consider the spin-1/2 transverse field Ising model on an infinite chain

$$H(h) = -\sum_{j=-\infty}^{\infty} \left[ \sigma_j^x \sigma_{j+1}^x + h\sigma_j^z \right]. \tag{1}$$

The ground state phase diagram features ferromagnetic ($h < 1$) and paramagnetic ($h > 1$) phases that are separated by a quantum critical point in the universality class of the two-dimensional Ising model [35]. The order parameter that characterizes the transition is the longitudinal magnetisation $\langle GS|\sigma_j^x|GS\rangle$. At finite temperature spontaneous breaking of the $\mathbb{Z}_2$ symmetry of $H(h)$ is forbidden and hence the order present in the ground state at $h < 1$ melts. In order for this paper to be self-contained we now briefly summarize the relevant steps for diagonalizing the Hamiltonian (1). A more detailed discussion can be found in e.g. the Appendix in [42]. The TFIC is mapped to a model of spinless fermions by a Jordan-Wigner transformation

$$\sigma_j^z = 1 - 2c_j^\dagger c_j \,, \qquad \sigma_j^x = \prod_{l=-\infty}^{j-1} (1 - 2c_l^\dagger c_l)(c_j + c_j^\dagger) \,, \tag{2}$$

where $c_j$ are fermion operators obeying canonical anticommutation relations $\{c_j^\dagger, c_k\} = \delta_{j,k}$. Setting aside the issue of boundary conditions the Hamiltonian takes the form

$$H(h) = -J \sum_{j=-\infty}^{\infty} (c_j^\dagger - c_j)(c_{j+1} + c_{j+1}^\dagger) - Jh(c_j c_j^\dagger - c_j^\dagger c_j). \tag{3}$$

This is diagonalized by a Bogoliubov transformation

$$c_j = \int_{-\pi}^{\pi} \frac{dk}{2\pi} e^{-ikj} \left[ \cos(\theta_k/2)\alpha_k + i \sin(\theta_k/2)\alpha_{-k}^\dagger \right], \tag{4}$$

where $\{\alpha_k, \alpha_p^\dagger\} = \delta_{p,k}$ and the Bogoliuobov angle is

$$e^{i\theta_k} = \frac{h - e^{ik}}{\sqrt{1 + h^2 - 2h\cos k}} . \tag{5}$$

The Hamiltonian takes the form

$$H(h) = \int_{-\pi}^{\pi} \frac{dk}{2\pi} \varepsilon(k) \left[ \alpha_k^\dagger \alpha_k - \frac{1}{2} \right], \tag{6}$$

where the dispersion relation is given by

$$\varepsilon(k) = 2J\sqrt{1 + h^2 - 2h\cos(k)}. \tag{7}$$

The ground state of $H(h)$ is equal to the Bogoliubov vacuum state defined by

$$\alpha_k |0\rangle = 0. \tag{8}$$

## 2.2 Full Counting Statistics and Generating Function

We are interested in the properties of the smooth and staggered components of the transverse magnetization of a chain segment of length $\ell$. These are defined as

$$S_u^z(\ell) = \sum_{j=1}^{\ell} \sigma_j^z , \qquad S_s^z(\ell) = \sum_{j=1}^{\ell} (-1)^j \sigma_j^z . \tag{9}$$

Given a density matrix $\rho$ that specifies the quantum mechanical state of our system, the probability distributions for the transverse subsystem magnetizations are given by

$$P^{(u,s)}(m) = \text{Tr}\left( \rho \, \delta\big(m - S_{u,s}^z(\ell)\big) \right) . \tag{10}$$

In the following we will focus on the characteristic functions of these probability distributions, defined as

$$\begin{aligned} P^{(u,s)}(m) &= \int_{-\infty}^{\infty} \frac{d\lambda}{2\pi} e^{-i\lambda m} \chi^{(u,s)}(\lambda, \ell) , \\ \chi^{(u,s)}(\lambda, \ell) &= \text{Tr}\left[ \rho \, e^{i\lambda S_{u,s}^z} \right]. \end{aligned} \tag{11}$$

By construction, the expansion of $\chi^{(u,s)}(\lambda, \ell)$ around $\lambda = 0$ generates the moments of the associated probability distribution. The following relations are readily inferred from the definition of $\chi^{(u,s)}(\lambda, \ell)$

$$\begin{aligned} \chi^{(u,s)}(\lambda, \ell) &= \left[ \chi^{(u,s)}(-\lambda, \ell) \right]^* , \\ \chi^{(u,s)}(0, \ell) &= 1 , \\ \chi^{(u,s)}(\lambda + \pi, \ell) &= (-1)^\ell \chi^{(u,s)}(\lambda, \ell) . \end{aligned} \tag{12}$$

These properties imply

$$P^{(\mathrm{u,s})}(m) = 2 \sum_{r \in \mathbb{Z}} P_w^{(\mathrm{u,s})}(r) \begin{cases} \delta(m - 2r + \ell) & \text{if } \ell \text{ is odd} \\ \delta(m - 2r) & \text{if } \ell \text{ is even} \end{cases} \tag{13}$$

where we have defined the weights

$$P_w^{(\mathrm{u,s})}(r) = \int_{-\pi/2}^{\pi/2} \frac{d\lambda}{2\pi} e^{-2i\lambda r} \chi^{(\mathrm{u,s})}(\lambda, \ell) . \tag{14}$$

## 3 Generating Function for a general Gaussian state

In this section we show how to obtain the generating function (11) for a general Gaussian state with a novel method that is however equivalent to the one used in [14].

Our starting point is the realization that (11) depends only on the reduced density matrix of the block $A$ of $\ell$ adjacent spins

$$\chi^{(\mathrm{u,s})}(\lambda, \ell) = \mathrm{Tr}\left[\rho \, e^{i\lambda S_{u,s}^z(\ell)}\right] = \mathrm{Tr}\left[\rho_A \, e^{i\lambda S_{u,s}^z(\ell)}\right] \equiv \widetilde{Z} \, \mathrm{Tr}\left[\rho_A \, \widetilde{\rho}^{(\mathrm{u,s})}\right], \quad a = u, s, \tag{15}$$

where we have introduced the auxiliary "density matrices"

$$\widetilde{\rho}^{(\mathrm{u,s})} \equiv \frac{1}{\widetilde{Z}^{(\mathrm{u,s})}} e^{i\lambda S_{u,s}^z(\ell)}, \qquad \widetilde{Z}^{(\mathrm{u,s})} = \mathrm{Tr}\left[e^{i\lambda S_{u,s}^z(\ell)}\right] = (2\cos(\lambda))^\ell . \tag{16}$$

Here the "partition function" $\widetilde{Z}^{(a)}$ ensures the normalisation $\mathrm{Tr}\left(\widetilde{\rho}^{(a)}\right) = 1$. A fundamental property that we will exploit in the following is that both $\rho_A$ and $\widetilde{\rho}^{(a)}$ are Gaussian operators in the fermionic representation of our problem, *cf.* section 2.1. Hence they are univocally determined by the correlation matrices of the fundamental fermionic operators [50–52]. Moreover, the trace of the product of Gaussian operators such as (15) can be expressed in terms of the associated correlation matrices [53]. This is a very useful property, see e.g. Ref. [47] for a related application, that forms the basis of our analysis.

In order to proceed we need to specify a convenient basis of operators. This is provided by Majorana fermions related to the lattice spin operators by

$$a_{2l-1} = \left(\prod_{m<l} \sigma_m^z\right)\sigma_l^x, \qquad a_{2l} = \left(\prod_{m<l} \sigma_m^z\right)\sigma_l^y, \qquad \sigma_l^z = i a_{2j} a_{2j-1}. \tag{17}$$

The Majorana fermions satisfy the algebra

$$\{a_j, a_k\} = 2\delta_{j,k} . \tag{18}$$

They are related to the Jordan-Wigner fermions (2) by $a_{2l-1} = c_l^\dagger + c_l$ and $a_{2l} = -i(c_l^\dagger - c_l)$.

As we are dealing with Gaussian density matrices we can follow Refs. [50–52] and Wick's theorem to express $\rho_A$ in terms of the subsystem *correlation matrix* $\Gamma_{nm}^A$

$$\Gamma_{nm}^A = \mathrm{Tr}[\rho \, a_m a_n] - \delta_{nm} , \qquad 1 \le m, n \le 2\ell. \tag{19}$$

As the Pauli matrices form a basis in the space of operators over $\mathbb{C}^2$ the reduced density matrix of a subsystem $A$ that consists of $\ell$ neighbouring spins at sites $i = 1, \ldots, \ell$ can be expressed in the form

$$\rho_A = \frac{1}{2^\ell} \sum_{\{\alpha_1 \ldots \alpha_\ell\}} \mathrm{Tr}\left(\rho \, \sigma_1^{\alpha_1} \ldots \sigma_\ell^{\alpha_\ell}\right) \sigma_1^{\alpha_1} \ldots \sigma_\ell^{\alpha_\ell} , \tag{20}$$

where $\alpha_i = 0, x, y, z$. We now restrict our discussion tor density matrices that are invariant under the $\mathbb{Z}_2$ transformation

$$P\sigma_l^z P = \sigma_l^z , \qquad P\sigma_l^{x,y} P = -\sigma_l^{x,y} . \tag{21}$$

In this case the Jordan-Wigner strings cancel and the reduced density matrix (RDM) is mapped to an operator expressed in terms of Majorana fermions acting on the same spatial domain

$$\rho_A = \frac{1}{2^\ell} \sum_{\{\mu_1 \dots \mu_\ell = 0,1\}} \mathrm{Tr}\left(\rho \, a_1^{\mu_1} \dots a_{2\ell}^{\mu_{2\ell}}\right) a_{2\ell}^{\mu_{2\ell}} \dots a_1^{\mu_1} . \tag{22}$$

We note that the case where $P\rho P \neq \rho$ can be dealt with by the method set out in Ref. [47]. The RDM (22) can be written in an explicit Gaussian form as

$$\rho_A = \frac{1}{Z}\exp\left[\frac{1}{4}\sum_{m,n} a_m W_{mn} a_n\right], \tag{23}$$

where $W$ is a skew symmetric $2\ell \times 2\ell$ hermitian matrix. Using Wick's theorem the matrix $W$ can be related to the correlation matrix (19)

$$\tanh\frac{W}{2} = \Gamma^A. \tag{24}$$

The auxiliary density matrices $\widetilde{\rho}^{(u,s)}$ (16) can be expressed in the Majorana basis in a completely analogous way. The corresponding $2\ell \times 2\ell$ correlation matrices $\widetilde{\Gamma}^{(u,s)}$ are given by

$$\widetilde{\Gamma}_{ij}^{(u)} = \mathrm{Tr}\left[\widetilde{\rho}^{(u)} a_j a_i\right] = \frac{1}{\widetilde{Z}^{(u)}}\mathrm{Tr}\left(\prod_{k=1}^{\ell}(\cos\lambda - i\sin\lambda a_{2k}a_{2k-1})a_j a_i\right) - \delta_{ij} ,$$

$$\widetilde{\Gamma}_{ij}^{(s)} = \mathrm{Tr}\left[\widetilde{\rho}^{(s)} a_j a_i\right] = \frac{1}{\widetilde{Z}^{(s)}}\mathrm{Tr}\left[\prod_{k=1}^{\ell}(\cos(\lambda) - i(-1)^k \sin(\lambda) a_{2k}a_{2k-1})a_j a_i\right] - \delta_{ij}. \tag{25}$$

The only non-vanishing matrix elements are

$$\widetilde{\Gamma}_{2j,2j-1}^{(u)} = -\widetilde{\Gamma}_{2j-1,2j}^{(u)} = \frac{1}{2\cos\lambda}\mathrm{Tr}\left[(\cos\lambda - i\sin\lambda \, a_{2j}a_{2j-1})a_{2j-1}a_{2j}\right] = -\tan\lambda ,$$

$$\widetilde{\Gamma}_{2j,2j-1}^{(s)} = -\widetilde{\Gamma}_{2j-1,2j}^{(s)} = \frac{1}{2\cos\lambda}\mathrm{Tr}\left[(\cos\lambda - i(-1)^j \sin\lambda \, a_{2j}a_{2j-1})a_{2j-1}a_{2j}\right]$$

$$= -(-1)^j \tan\lambda . \tag{26}$$

This implies that $\widetilde{\Gamma}^{(u,s)}$ are block-diagonal, e.g.

$$\widetilde{\Gamma}^{(u)} = i\tan\lambda \begin{bmatrix} \sigma_y & 0 & \dots \\ 0 & \sigma_y & \dots \\ & & \ddots \end{bmatrix} \equiv i\tan(\lambda)\Sigma_y, \tag{27}$$

where $\sigma_y$ is the $2 \times 2$ Pauli matrix.

We are now in a position to write down a convenient determinant representation for the generating functions $\chi^{(u,s)}(\lambda,\ell)$. To do so we employ a relation derived in Ref. [53]: given two Gaussian density matrices $\rho_{1,2}$ with correlation matrices $\Gamma_{1,2}$ the trace of their product is given by

$$\mathrm{Tr}[\rho_1 \rho_2] = \sqrt{\det\left(\frac{1 + \Gamma_1\Gamma_2}{2}\right)}. \tag{28}$$

Applying this relation to our case we arrive at the following determinant representations

$$\chi^{(a)}(\lambda,\ell) = \frac{1}{(2\cos\lambda)^\ell}\sqrt{\det\left(\frac{1+\Gamma^A\widetilde{\Gamma}^{(a)}}{2}\right)}, \quad a = u,s, \tag{29}$$

where $\Gamma^A$ and $\Gamma^{(u,s)}$ are given in (19) and (26), (27) respectively.

## 3.1 Simplification in special cases

Equation (29) has been derived for a general $\mathbb{Z}_2$-invariant Gaussian state with density matrix $\rho$. If the state is also invariant under translations and reflections with respect to a site the generating function $\chi^{(u)}(\lambda,\ell)$ can be simplified further. Indeed, under these conditions, the correlation matrix assumes a block Toeplitz form [45,51]

$$\Gamma^A = \begin{pmatrix} \Pi_0 & \Pi_{-1} & \dots & \Pi_{1-\ell} \\ \Pi_1 & \Pi_0 & & \vdots \\ \vdots & & \ddots & \vdots \\ \Pi_{\ell-1} & \dots & \dots & \Pi_0 \end{pmatrix}, \qquad \Pi_l = \begin{pmatrix} -f_l & g_l \\ -g_{-l} & f_l \end{pmatrix}, \tag{30}$$

where

$$\begin{aligned} g_l &= \mathrm{Tr}\big(a_{2n}a_{2n+2l-1}\big) = -\mathrm{Tr}\big(a_{2n-1}a_{2n-2l}\big), \\ f_l &= \mathrm{Tr}\big(a_{2n}a_{2n+2l}\big) - \delta_{l0}. \end{aligned} \tag{31}$$

Taking advantage of the block diagonal form of the correlation matrix of the auxiliary density matrix in (27) we can cast the generating function in the form

$$\chi^{(u)}(\lambda,\ell) = (2\cos\lambda)^\ell\sqrt{\det\left(\frac{1-\tan(\lambda)\Gamma'}{2}\right)}, \tag{32}$$

where $\Gamma'$ is a block Toeplitz matrix

$$\Gamma' = \begin{pmatrix} \Pi'_0 & \Pi'_{-1} & \dots & \Pi'_{1-\ell} \\ \Pi'_1 & \Pi'_0 & & \vdots \\ \vdots & & \ddots & \vdots \\ \Pi'_{\ell-1} & \dots & \dots & \Pi'_0 \end{pmatrix}, \qquad \Pi'_l = \begin{pmatrix} g_l & f_l \\ f_l & g_{-l} \end{pmatrix}. \tag{33}$$

## 3.2 Expressions for the first few cumulants

The determinant representation (29) of the generating function provides an efficient way for determining the cumulants of the probability distribution, which is the main purpose of the function itself. The cumulants are obtained in the usual way from the series expansions of $\ln\chi^{(u,s)}(\lambda,\ell)$

$$\ln\chi^{(u,s)}(\lambda,\ell) = \sum_{n=1}^{\infty} \frac{C_n^{(u,s)}}{n!}(i\lambda)^n. \tag{34}$$

The first few terms of the series expansion are

$$\begin{aligned} \ln\chi^{(u)}(\lambda,\ell) &= \ell\ln(\cos\lambda) - \frac{1}{2}\sum_{n=1}^{\infty}\frac{(\tan\lambda)^n}{n}\mathrm{Tr}\big[(\bar{\Gamma})^n\big] \\ &= -\ell\frac{\lambda^2}{2} - \frac{\lambda}{2}\mathrm{Tr}\big(\bar{\Gamma}\big) - \frac{\lambda^2}{4}\mathrm{Tr}\big(\bar{\Gamma}^2\big) - \frac{\lambda^3}{6}\big(\mathrm{Tr}\big(\bar{\Gamma}^3\big) + \mathrm{Tr}\big(\bar{\Gamma}\big)\big) + O(\lambda^4), \end{aligned} \tag{35}$$

where we have defined

$$\bar{\Gamma} = -i\Gamma_A \Sigma_y \ . \tag{36}$$

The first three cumulants are

$$C_1 = \frac{i}{2}\mathrm{Tr}(\bar{\Gamma}) \ , \quad C_2 = \ell + \frac{\mathrm{Tr}(\bar{\Gamma}^2)}{2} \ , \quad C_3 = -\frac{i}{2}(\mathrm{Tr}(\bar{\Gamma}^3) + \mathrm{Tr}(\bar{\Gamma}))). \tag{37}$$

Specifying to the case of density matrices $\rho$ that are invariant under translations and reflections around a site we have

$$\mathrm{Tr}(\bar{\Gamma}) = \ell\,\mathrm{Tr}(\Pi'_0) = 2\ell g_0, \tag{38}$$

$$\mathrm{Tr}(\bar{\Gamma}^2) = \ell\,\mathrm{Tr}\left(\sum_{j=0}^{\ell-1}(2(\ell-j)-\ell\delta_{j0})\Pi'_j \Pi'_{-j}\right) = 2\ell\sum_{j=0}^{\ell-1}(2(\ell-j)-\ell\delta_{j0})(g_j g_{-j} + f_j f_{-j})$$

$$= 2\ell\left(\sum_{j=1}^{\ell-1}(2(\ell-j))(g_j g_{-j} + f_j f_{-j}) + \ell g_0^2\right) = 2\mathrm{Tr}(F^2 + G^2), \tag{39}$$

$$\mathrm{Tr}(\bar{\Gamma}^3) = 2\mathrm{Tr}(G^3 + 3F^2 G), \tag{40}$$

where $F$ and $G$ are the $\ell \times \ell$ Toeplitz matrices

$$G = \begin{pmatrix} g_0 & g_{-1} & \cdots & g_{1-\ell} \\ g_1 & g_0 & & \vdots \\ \vdots & & \ddots & \vdots \\ g_{\ell-1} & \cdots & \cdots & g_0 \end{pmatrix}, \qquad F = \begin{pmatrix} f_0 & f_{-1} & \cdots & f_{1-\ell} \\ f_1 & f_0 & & \vdots \\ \vdots & & \ddots & \vdots \\ f_{\ell-1} & \cdots & \cdots & f_0 \end{pmatrix}. \tag{41}$$

For the first three cumulants we obtain

$$C_1 = i\ell g_0 \ , \quad C_2 = \ell + \mathrm{Tr}(F^2 + G^2) \ , \quad C_3 = -i(\mathrm{Tr}(G^3 + 3F^2 G) + \ell g_0) \ . \tag{42}$$

It is straightforward to generalise these considerations to higher cumulants because $\mathrm{Tr}(\bar{\Gamma}^n)$ can always be written as the sum of the traces of products of $F$ and $G$.

# 4 Full counting statistics in equilibrium

In this section we analyze the generating function $\chi^{(u,s)}(\lambda, \ell)$ obtained from (29) and the associated probability distribution in equilibrium configurations. We first consider the ground state FCS, which has been previously studied by Cherng and Demler in [9]. We then turn to the FCS in finite temperature equilibrium states, which to the best of our knowledge has not been considered in the literature.

## 4.1 Full counting statistics in the ground state

In the ground state the generating function is of the form (32), (33) with entries

$$f_l = 0, \tag{43}$$

$$g_l = -i\int_{-\pi}^{\pi} \frac{dk}{2\pi} e^{-ikl} e^{i\theta_k} \ , \tag{44}$$

where $\theta_k$ is the Bogoliubov angle (5). By rearranging rows and columns, $\bar{\Gamma}$ can be brought to a block diagonal form with $\ell \times \ell$ matrices $G$ and $G^T$ (41) on the diagonal and zero otherwise. This allows us to express the generating function as

$$\chi^{(u)}(\lambda, \ell) = (2\cos\lambda)^\ell \sqrt{\det\left(\frac{1-\tan(\lambda)G}{2}\right)\det\left(\frac{1-\tan(\lambda)G^T}{2}\right)} = \det(\cos\lambda - \sin(\lambda)G),$$
(45)

This is precisely the result previously obtained by Cherng and Demler [9] by a different technique. They considered the generating function

$$\chi_{CD}(\lambda, \ell) = \langle GS|e^{i\lambda \sum_{j=1}^\ell \frac{1-\sigma_j^z}{2}}|GS\rangle = e^{i\lambda\ell/2}\chi^{(u)}(-\lambda/2, \ell) = \det\left(\frac{1+e^{i\lambda}}{2} + \frac{1-e^{i\lambda}}{2}iG\right).$$
(46)

The Toeplitz determinant (45) can be analyzed by standard methods [9]. The *symbol* $\tau(e^{ik})$ of a block Toeplitz $T_\ell$ with elements $(T_\ell)_{ln} = t_{l-n}$ is defined through the equation

$$t_n = \int_0^{2\pi} \frac{dk}{2\pi}\tau(e^{ik})e^{-ink}.$$
(47)

The symbol of the block Toeplitz matrix (45) is given by

$$\tau(e^{ik}) = \cos\lambda + ie^{i\theta_k}\sin\lambda.$$
(48)

As long as the symbol has zero winding number a straightforward application of Szegő's Lemma gives, *cf.* Appendix A

$$\lim_{\ell \to \infty} \frac{\ln\chi^{(u)}(\lambda, \ell)}{\ell} = \int_0^{2\pi} \frac{dk}{2\pi}\ln(\cos\lambda + ie^{i\theta_k}\sin\lambda).$$
(49)

For $h < 1$ and $\lambda > \lambda_c(h)$ the winding number of the symbol is 1 and the above result gets modified accordingly [11]. For a detailed analysis we refer to Ref. [11]. The full counting statistics for the transverse magnetization in the entire system was studied in [54] and the result is identical to (49). Thus considering the subsystem instead of the entire system only makes a difference for $h < 1$ and $\lambda > \lambda_c(h)$, as discussed in [9].

We note that all cumulants can be obtained from (49) since they are defined by the expansion close to $\lambda = 0$. Consequently, the first cumulants are given by

$$\begin{aligned}
C_1 &= \int_{-\pi}^{\pi} \frac{dk}{2\pi}e^{i\theta_k}, \\
C_2 &= \int_{-\pi}^{\pi} \frac{dk}{2\pi}(1 - e^{2i\theta_k}), \\
C_3 &= \int_{-\pi}^{\pi} \frac{dk}{2\pi}2(e^{3i\theta_k} - e^{i\theta_k}), \\
C_4 &= \int_{-\pi}^{\pi} \frac{dk}{2\pi}2(-1 + 4e^{2i\theta_k} - 3e^{4i\theta_k}).
\end{aligned}$$
(50)

The non-zero values of $C_3$ and $C_4$ show that the probability distribution is non-Gaussian.

## 4.2 Full counting statistics at finite temperature

We now turn to the FCS in finite temperature equilibrium states, for which we are not aware of any results in the literature. In this case, the correlation matrix has the same structure as for the ground state, but now

$$f_l = 0, \tag{51}$$

$$g_l = -i \int_{-\pi}^{\pi} \frac{dk}{2\pi} e^{-ikl} e^{i\theta_k} \tanh(\beta \epsilon_k / 2), \tag{52}$$

where $\epsilon_k$ is the dispersion relation (7). Since $f_l = 0$, the same simplifications as in the ground state case apply and the generating function can be expressed as

$$\chi^{(u)}(\lambda, \ell) = \det(\cos \lambda - \sin(\lambda) G). \tag{53}$$

In Fig. 1 we show $P_w^{(u)}(m)$ for subsystem size $\ell = 20$ and several different temperatures. We

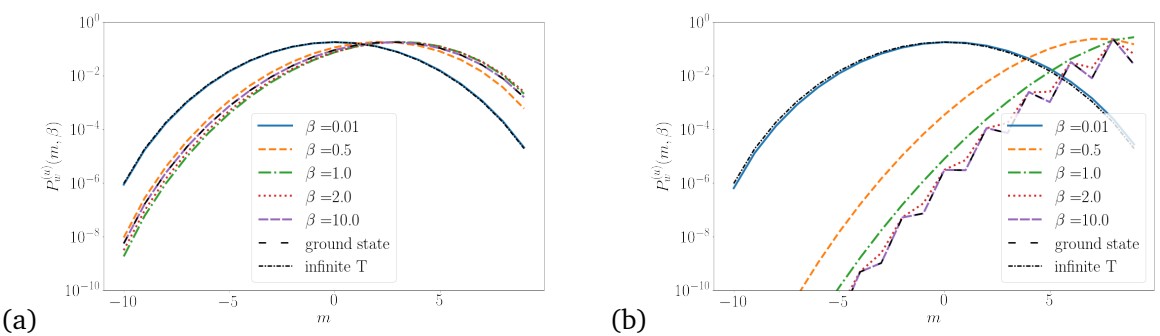

Figure 1: Probability distribution as a function of $m$ for $\ell = 20$ and several temperatures at (a) $h = 0.5$; (b) $h = 2$.

employ a log-linear plot in order to make the deviations of the probability distributions from a Gaussian form (which would correspond to a parabolic form) more apparent. We can see from Fig. 1 (a) that the temperature dependence for $h < 1$, corresponding to the ferromagnetically ordered phase at zero temperature, is not very pronounced. In contrast we see a much stronger temperature dependence in the paramagnetic phase, *cf.* Fig. 1 (b). At low temperatures the probability distribution is as expected asymmetric as a result of the applied field and is seen to display an even/odd structure. The latter disappears quickly as temperature is increased, whereas the asymmetry remains until the temperature exceeds the scale set by the magnetic field.

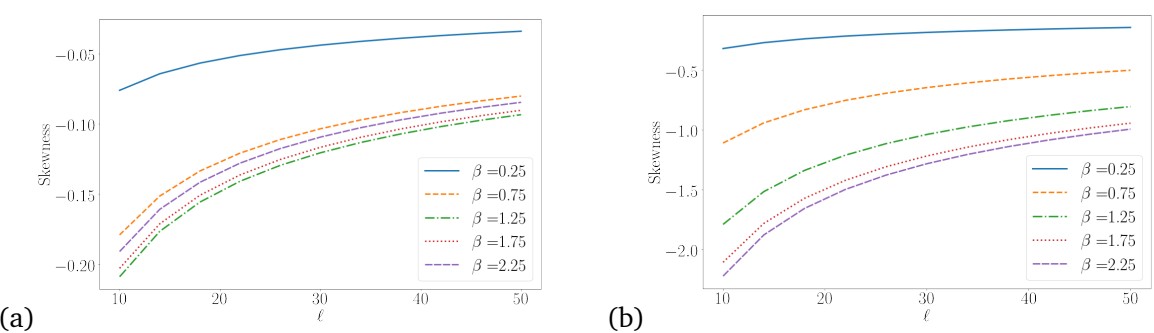

Figure 2: Skewness as a function of $\ell$ for several values of $\beta$ at (a) $h = 0.5$ and (b) $h = 2$.

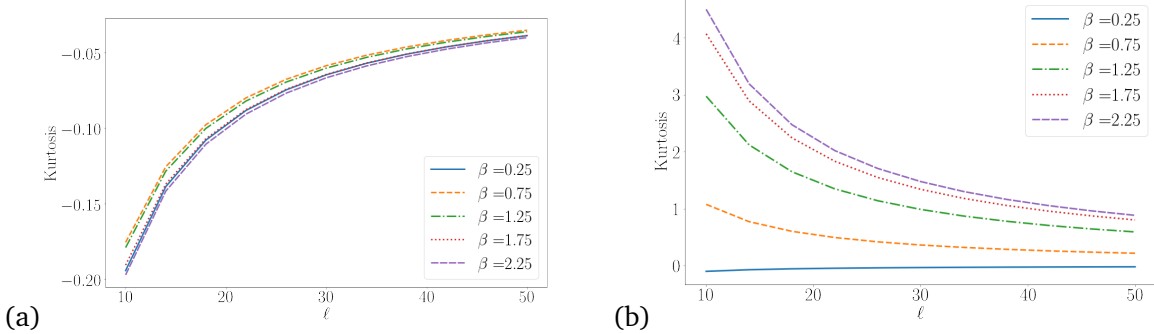

Figure 3: Excess kurtosis as a function of subsystem size $\ell$ for several temperatures and (a) $h = 0.5$ and (b) $h = 2$.

In Figs 2 and 3 we show the skewness and excess kurtosis of the probability distribution as a function of subsystem size $\ell$ for a range of temperatures. These are defined as the thermal expectation values

$$\left\langle \left[ \frac{X}{\sqrt{\langle X^2 \rangle_\beta}} \right]^3 \right\rangle_\beta , \quad \left\langle \left[ \frac{X}{\sqrt{\langle X^2 \rangle_\beta}} \right]^4 \right\rangle_\beta - 3 , \quad X = S_u^z(\ell) - \langle S_u^z(\ell) \rangle_\beta . \tag{54}$$

Both skewness and excess kurtosis are non-vanishing for finite $\beta$ and $\ell$, which establishes that the distribution is not Gaussian. A very peculiar feature is that at fixed $\ell$ skewness and excess kurtosis are non-monotonic functions of the temperature. Furthermore, we observe that at a fixed temperature they both tend to zero as the subsystem size $\ell$ is increased. This signals that the corresponding probability distribution approaches a Gaussian. This is expected as for large subsystem sizes the laws of thermodynamics apply and the probability distribution is then approximately Gaussian with a standard deviation that scales as $\sqrt{\ell}$.

## 5 Full counting statistics after a quantum quench

We now turn to the time evolution of the characteristic function $\chi^{(u,s)}(\lambda, t)$ after quantum quenches. We consider two different classes of initial states:

- We initialize the system in the ground state of $H(h_0)$ and time evolve with $H(h)$. Such transverse field quenches have been studied in detail in the literature [40–47, 55–65].

- We initialize the system in the Néel state $|\uparrow\downarrow\uparrow\downarrow \ldots \uparrow\downarrow\rangle$, thus breaking translational symmetry by one site. This symmetry is restored at late times after the quench and it is an interesting question how this is reflected in the probability distributions of observables.

### 5.1 Transverse field quench $h_0 \longrightarrow h$

In this quench protocol both the Hamiltonian and the initial state are translationally invariant. The characteristic function has the determinant representation (32), (33) with [45]

$$g_l = -i \int_{-\pi}^{\pi} \frac{dk}{2\pi} e^{-ikl} e^{i\theta_k} \left( \cos \Delta_k - i \sin \Delta_k \cos(2\varepsilon_k t) \right) \tag{55}$$

$$f_l = \int_{-\pi}^{\pi} \frac{dk}{2\pi} e^{-ikl} \sin \Delta_k \sin(2\varepsilon_k t) , \tag{56}$$

where

$$e^{i\theta_k} = \frac{h - e^{ik}}{\sqrt{1 + h^2 - 2h\cos k}} , \quad \cos\Delta_k = 4\frac{hh_0 - (h + h_0)\cos k + 1}{\varepsilon_h(k)\varepsilon_{h_0}(k)} . \tag{57}$$

Using Szegő's Lemma it is straightforward to obtain the large-$\ell$ asymptotics in the initial ($t = 0$) and stationary ($t = \infty$) states. The $t = 0$ result corresponds to a ground state at field $h_0$ and has been discussed earlier.

### 5.1.1 Behaviour in the stationary state

The late time asymptotics of the generating function can be determined from Szegő's Lemma. For quenches into the paramagnetic phase $h > 1$ it takes the form

$$\lim_{t\to\infty} \frac{\ln\chi^{(u)}(\lambda,\ell,t)}{\ell} = \int_0^{2\pi} \frac{dk}{2\pi} \ln\left(\cos\lambda + i\sin\lambda\cos\Delta_k e^{i\theta_k}\right) + \mathcal{O}(1/\ell) , \quad \ell \gg 1. \tag{58}$$

The $\mathcal{O}(\ell^{-1})$ corrections also follow from Szegő's Lemma. The real and imaginary parts of

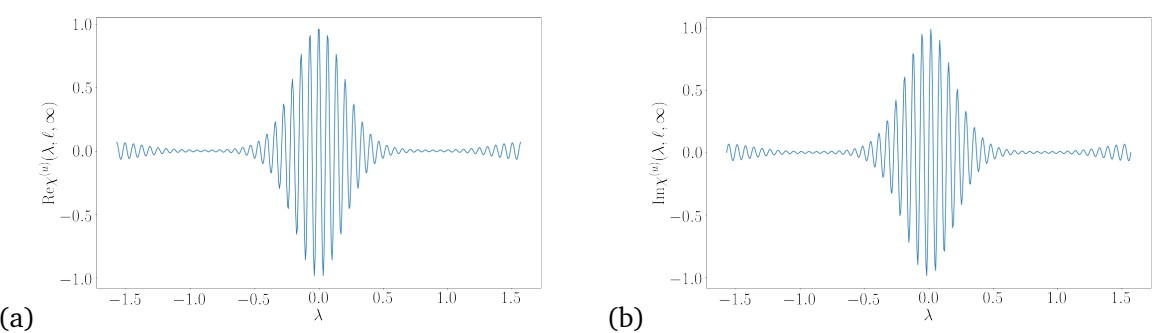

(a)                                              (b)

Figure 4: (a) $\mathrm{Re}\chi^{(u)}(\lambda,\ell,\infty)$ and $\mathrm{Im}\chi^{(u)}(\lambda,\ell,\infty)$ for a quench from $h = 5$ to $h = 2$ and subsystem size $\ell = 100$.

$\chi^{(u)}(\lambda,\ell,t)$ (with $\mathcal{O}(\ell^{-1})$ corrections included) are shown for a transverse field quench from $h_0 = 5$ to $h = 2$ and subsystem size $\ell = 100$ in Fig. 4.

For quenches into the ferromagnetic phase and $\lambda < \lambda_c(h_0,h)$, Eq. (58) continues to hold. However, for $\lambda > \lambda_c(h_0,h)$ the symbol exhibits non-zero winding number and the analysis needs to be modified, *cf.* Appendix A. The probability distribution in the stationary state is obtained by Fourier transforming $\chi^{(u)}(\lambda,\ell,t)$. Examples for several transverse field quenches are shown in Fig. 5. We again employ a logarithmic scale to make the deviations from a Gaussian form more apparent. In Figs 6 we plot the skewness and the excess kurtosis of the steady state probability distributions for a number of transverse field quenches. We observe that in all cases both skewness and excess kurtosis tend to zero for large subsystem sizes. This signals that the probability distributions approach Gaussians in the large-$\ell$ limit. While the steady states are non-thermal now, they still exhibit finite correlation lengths. Employing the same arguments as for finite temperature ensembles then implies that the cumulants of $S_u^z(\ell)$ are proportional to $\ell$ in the large-$\ell$ limit. This in turn suggests that skewness and excess kurtosis should scale as $\ell^{-1/2}$ and $\ell^{-1}$ respectively, while the standard deviation scales as $\ell^{1/2}$. These expectations are in perfect agreement with our findings.

### 5.1.2 Scaling collapse

At finite times the FCS and the probability distribution can be computed efficiently from the determinant representation (32). Importantly we observe that for sufficiently large values of

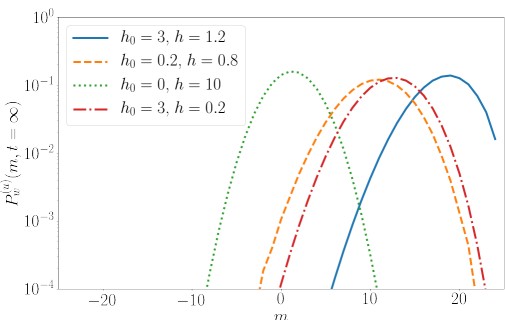

Figure 5: Stationary state probability distribution $P_w^{(u)}(m, \infty)$ for a subsystem of size $\ell = 70$ for several transverse field quenches.

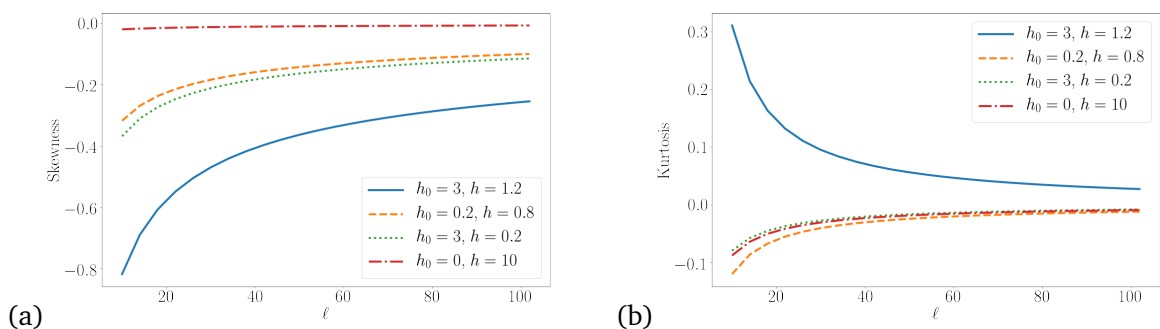

(a)  (b)

Figure 6: (a) Skewness and (b) Excess kurtosis of the steady state probability distribution as functions of subsystem size $\ell$ for a number of transverse field quenches.

$\ell$ and $t$ there is scaling collapse

$$\chi^{(u)}(\lambda, \ell, t) \approx \exp\left(\ell f(t/\ell)\right), \quad t, \ell \gg 1. \tag{59}$$

The property (59) is an important ingredient in the analytic calculation of the FCS described in section 6. Several examples of the scaling behaviour of the real part of the generating function are shown in Figs 7, 8. The imaginary parts exhibit a similar scaling collapse.

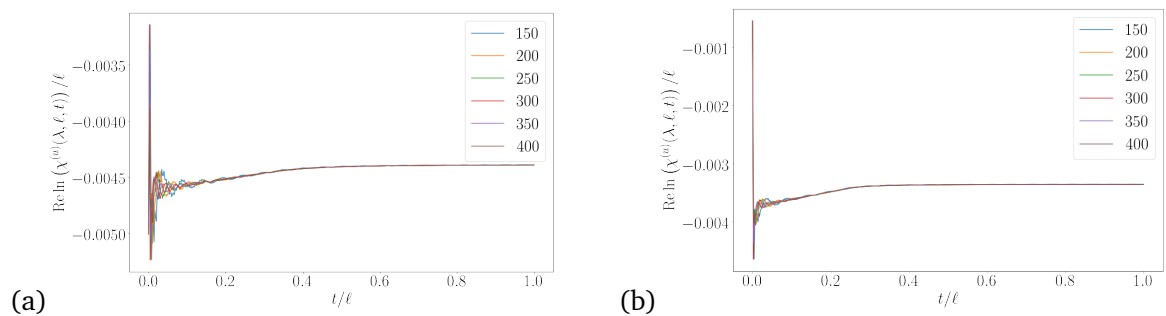

(a)  (b)

Figure 7: $\operatorname{Re} \ln \chi^{(u)}(0.1, \ell, t)/\ell$ for several values of $\ell$ for a quench from (a) $h = 0.2$ to $h = 0.8$ and (b) $h = 3$ to $h = 1.2$. The data for different subsystem sizes are seen to collapse at sufficiently late times.

For quenches *towards* the ferromagnetic regime the scaling collapse for general values of $\lambda$ can be significantly worse, and then really only emerges at rather large subsystem sizes $\ell$, *cf.* Fig. 9. Like in the case of the stationary state discussed above, *c.f.* section 5.1.1, there exists a critical value $\hat{\lambda}_c(h_0, h)$ of the counting parameter such that for $\lambda < \hat{\lambda}_c(h_0, h)$ the scaling

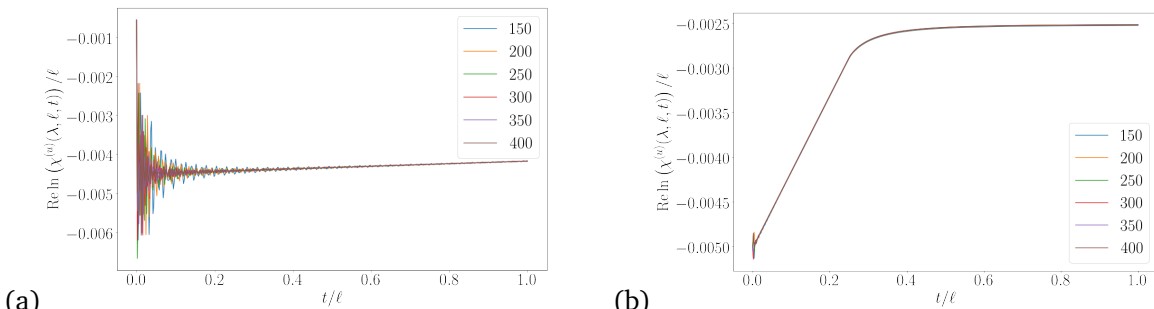

(a)  (b)

Figure 8: $\mathrm{Re}\ln\chi^{(u)}(0.1,\ell,t)/\ell$ for several values of $\ell$ for a quench from (a) $h=3$ to $h=0.2$ and (b) $h=0$ to $h=20$. The data for different subsystem sizes are seen to collapse at sufficiently late times.

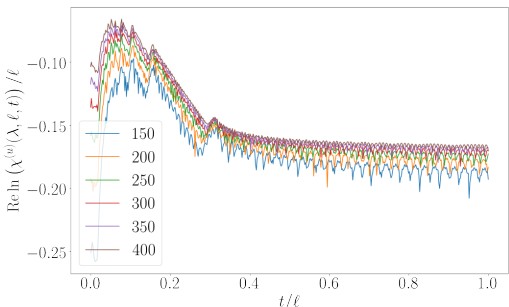

Figure 9: $\mathrm{Re}\ln\chi^{(u)}(1.4,\ell,t)/\ell$ for several values of $\ell$ for a quench from $h=0.2$ to $h=0.8$. The data for different subsystem sizes are seen to collapse at sufficiently late times only for very large subsystem sizes $\ell$.

collapse is excellent, while for $\lambda>\hat{\lambda}_c(h_0,h)$ no collapse is observed at the times and subsystem sizes of interest here. For the cases we have considered $\hat{\lambda}_c(h_0,h)$ coincides with $\lambda_c(h_0,h)$, which is the value of the counting parameter above which the symbol has non-zero winding number. We note however, that in cases like the one shown in Fig. 9 the generating function itself is extremely small and will not give a significant contribution to the corresponding probability distribution.

### 5.1.3 Time dependence of the probability distribution

There are basically four different kinds of transverse field quenches and we now consider them in turn.

1. Quenches within the ferromagnetic phase. For such quenches the probability distribution remains very narrow and approximately Gaussian throughout, *cf.* Fig. 10. For the parameters considered the average relaxes quickly towards its stationary value.

2. Quenches within the paramagnetic phase.

   Here the initial probability distribution exhibits an even/odd structure. This can be understood by doing perturbation theory around the large $h_0$ limit, *cf.* Appendix B. After the quench the mean of the probability distribution broadens and shifts towards smaller values of $m$. The alternating structure is initially preserved but then gets smoothed out. At late times $P_w^{(u)}(m,t)$ is well described by a Gaussian.

3. Quenches from the paramagnetic to the ferromagnetic phase.

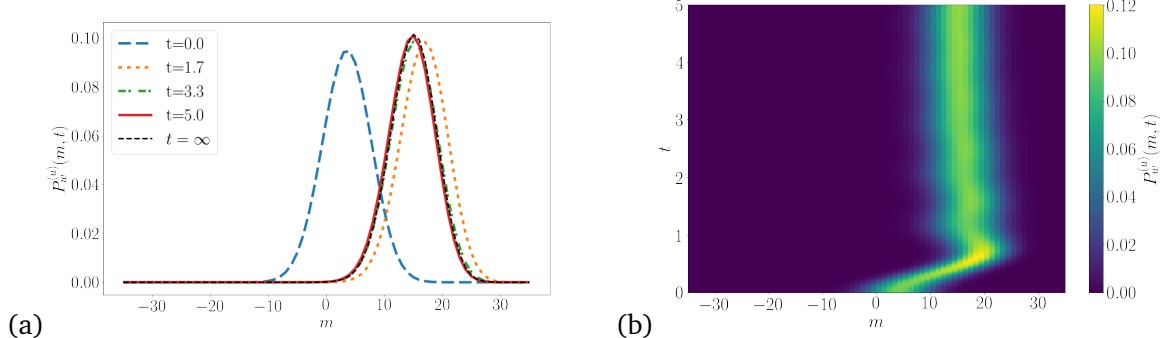

(a)                                                                                (b)

Figure 10: (a) Probability distribution $P_w^{(u)}(m,t)$ at times $t = 0, 1.7, 3.3, 5.0$ after a quench from $h = 0.2$ to $h = 0.8$ for subsystem size $\ell = 70$. (b) Probability distribution $P_w^{(u)}(m,t)$ for the same parameters.

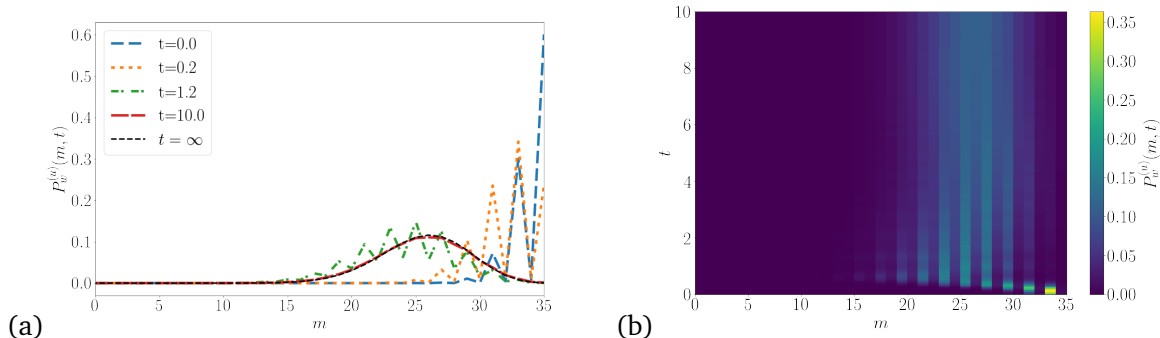

(a)                                                                                (b)

Figure 11: (a) Probability distribution $P_w^{(u)}(m,t)$ at times $t = 0, 0.2, 1.2, 10.0$ after a quench from $h = 3$ to $h = 1.2$ for subsystem size $\ell = 70$. (b) Probability distribution $P_w^{(u)}(m,t)$ for the same parameters.

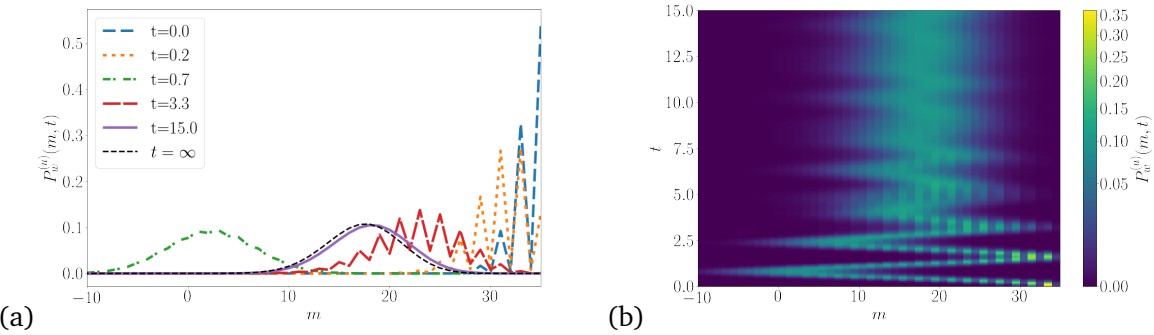

(a)                                                                                (b)

Figure 12: (a) Probability distribution $P_w^{(u)}(m,t)$ at times $t = 0, 0.2, 0.7, 3.3, 15.0$ after a quench from $h = 3$ to $h = 0.2$ for subsystem size $\ell = 70$. (b) Probability distribution $P_w^{(u)}(m,t)$ for the same parameters.

Here the probability distribution is initially peaked at a large value of $m$ and displays an even/odd structure. At later times it broadens and becomes smooth, while relaxing towards its stationary profile in an strongly oscillatory manner.

4. Quenches from the ferromagnetic to the paramagnetic phase.

In this case the probability distribution shows very little variation in time and remains narrow and approximately Gaussian throughout the evolution. It was pointed out in Ref. [62] that the return amplitude exhibits a non-analyticity at some finite time $t^*$ after

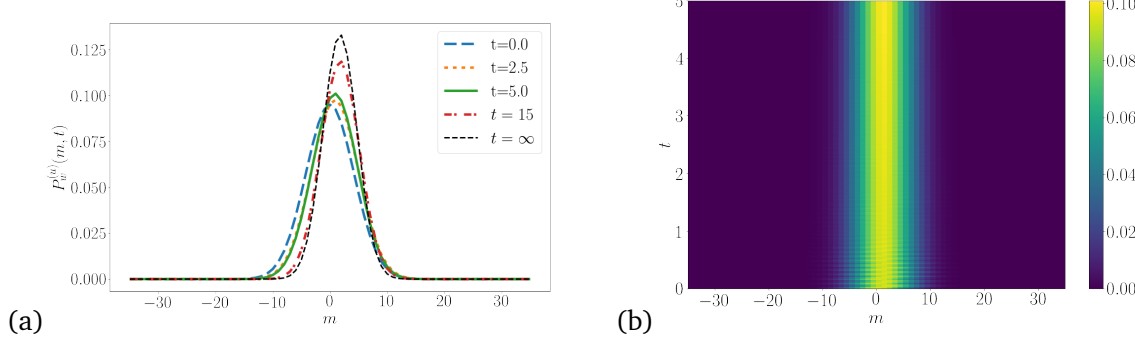

Figure 13: (a) Probability distribution $P_w^{(u)}(m,t)$ at times $t = 0, 2.5, 5.0$ after a quench from $h = 0$ to $h = 10$ for subsystem size $\ell = 70$. (b) Probability distribution $P_w^{(u)}(m,t)$ for the same parameters.

the quantum quench. This phenomenon was termed a "dynamical phase transition". Local operators are known to be insensitive to this phenomenon [41–43]. We have investigated the behaviour of $P_w^{(u)}(m,t)$ in the vicinity of $t^*$ but have not observed any unusual effects. We conclude that the probability distribution for the smooth subsystem magnetization in the transverse field direction is also insensitive to the "dynamical phase transition".

## 5.2 Quench from the Néel state

We now turn to the time evolution of $P_w^{(u,s)}(m,t)$ when the system is initialized in the Néel state $|\psi_0\rangle = |\uparrow\downarrow\uparrow\downarrow\ldots\uparrow\downarrow\rangle$. This explicitly breaks translational invariance by one site, but retains invariance under translation by two sites. As a result the subsystem correlation matrix is now a $4 \times 4$ block-Toeplitz matrix

$$
\Gamma_{\text{Néel}}^A = \begin{pmatrix} \Pi_0^{\text{Néel}} & \Pi_{-1}^{\text{Néel}} & \cdots & \Pi_{1-\ell/2}^{\text{Néel}} \\ \Pi_1^{\text{Néel}} & \Pi_0^{\text{Néel}} & & \vdots \\ \vdots & & \ddots & \vdots \\ \Pi_{\ell/2-1}^{\text{Néel}} & \cdots & \cdots & \Pi_0^{\text{Néel}} \end{pmatrix},
\tag{60}
$$

where we have assumed the subsystem size $\ell$ to be even and

$$
\begin{aligned}
\Pi_l^{\text{Néel}} &= \begin{pmatrix} \langle a_1 a_{4l+1}\rangle - \delta_{0l} & \langle a_2 a_{4l+1}\rangle & \langle a_3 a_{4l+1}\rangle & \langle a_0 a_{4l-3}\rangle \\ \langle a_1 a_{4l+2}\rangle & \langle a_2 a_{4l+2}\rangle - \delta_{l0} & \langle a_3 a_{4l+2}\rangle & \langle a_0 a_{4l-2}\rangle \\ \langle a_1 a_{4l+3}\rangle & \langle a_2 a_{4l+3}\rangle & \langle a_3 a_{4l+3}\rangle - \delta_{l0} & \langle a_0 a_{4l-1}\rangle \\ \langle a_1 a_{4l+4}\rangle & \langle a_2 a_{4l+4}\rangle & \langle a_3 a_{4l+4}\rangle & \langle a_0 a_{4l}\rangle - \delta_{l0} \end{pmatrix} \\
&= \begin{pmatrix} -f_l & g_l & h_l & 0 \\ -g_{-l} & f_l & 0 & h_l \\ -h_{-l} & 0 & f_l & -g_l \\ 0 & -h_{-l} & g_{-l} & -f_l \end{pmatrix} - \delta_{l,0}\mathbb{1}.
\end{aligned}
\tag{61}
$$

Here the various two point functions are given by

$$
\begin{aligned}
f_l - \delta_{l0} &= i \int_0^{2\pi} \frac{dk}{2\pi} e^{-2ijk} \Big[ e^{i\theta_k} \cos\big(\varepsilon(k+\pi)t\big) \sin\big(\varepsilon(k)t\big) \dots \\
&\qquad \dots - e^{-i\theta_{k+\pi}} \cos\big(\varepsilon(k)t\big) \sin\big(\varepsilon(k+\pi)t\big) \Big], \\
g_l &= i \int_0^{2\pi} \frac{dk}{2\pi} e^{-2ijk} \Big[ \cos\big(\varepsilon(k)t\big) \cos\big(\varepsilon(k+\pi)t\big) \dots \\
&\qquad \dots + e^{i(\theta_k + \theta_{k+\pi})} \sin\big(\varepsilon(k)t\big) \sin\big(\varepsilon(k+\pi)t\big) \Big], \\
h_l &= i \int_0^{2\pi} \frac{dk}{2\pi} e^{-i(2j-1)k} \Big[ e^{-i\theta_k} \cos\big(\varepsilon(k+\pi)t\big) \sin\big(\varepsilon(k)t\big) \dots \\
&\qquad \dots - e^{i\theta_{k+\pi}} \cos\big(\varepsilon(k)t\big) \sin\big(\varepsilon(k+\pi)t\big) \Big].
\end{aligned}
\tag{62}
$$

In the following we will determine the characteristic functions

$$
\chi^{(u)}(\lambda, \ell, t) = \langle \psi_0(t) | e^{i\lambda S_u^z(\ell)} | \psi_0(t) \rangle, \qquad \chi^{(s)}(\lambda, \ell, t) = \langle \psi_0(t) | e^{i\lambda S_s^z(\ell)} | \psi_0(t) \rangle,
\tag{63}
$$

where again we have defined

$$
S_u^z(\ell) = \sum_{j=1}^{\ell} \sigma_j^z, \quad S_s^z(\ell) = \sum_{j=1}^{\ell} (-1)^j \sigma_j^z.
\tag{64}
$$

According to our general discussion in section 3 they have determinant representations of the form

$$
\begin{aligned}
\chi^{(u)}(\lambda, \ell, t) &= \big(2\cos(\lambda)\big)^\ell \sqrt{\det\Big(\frac{1 + \Gamma_{\text{Néel}}^A \tilde{\Gamma}}{2}\Big)}, \\
\chi^{(s)}(\lambda, \ell, t) &= \big(2\cos(\lambda)\big)^\ell \sqrt{\det\Big(\frac{1 + \Gamma_{\text{Néel}}^A \tilde{\Gamma}^\pi}{2}\Big)},
\end{aligned}
\tag{65}
$$

where $\tilde{\Gamma}_{2j,2j-1}^\pi = -\tilde{\Gamma}_{2j-1,2j}^\pi = -\tan(\lambda)(-1)^j$ and $\tilde{\Gamma}_{2j,2j-1} = -\tilde{\Gamma}_{2j-1,2j} = -\tan(\lambda)$ respectively.

### 5.2.1 Behaviour in the stationary state

We first consider the probability distributions for a finite subsystem of even size $\ell$ in the late time limit. As we will now show, the stationary state for the Néel quench is locally equivalent to an infinite temperature state. To see this we first note that the energy of the Néel state is

$$
\langle \psi_0 | H(h) | \psi_0 \rangle = 0.
\tag{66}
$$

It is easy to see using their explicit representation in terms of spins [49] that the expectation values of all higher conservation laws also vanish

$$
\langle \psi_0 | I^{(n,\pm)} | \psi_0 \rangle = 0.
\tag{67}
$$

This in turn implies that the conserved Bogoliubov mode occupation numbers are given by

$$
\langle \psi_0 | \alpha_k^\dagger \alpha_k | \psi_0 \rangle = \frac{1}{2}.
\tag{68}
$$

These characterize an infinite temperature equilibrium state. We conclude that the system will relax locally [49] to an infinite temperature steady state at late times after the quench. Using this observation it is then straightforward to work out the probability distributions $P^{(u)}(m, t = \infty)$ of $S_u^z(\ell) = \sum_{j=1}^{\ell} \sigma_j^z$ and $P^{(s)}(m, \infty)$ of $S_s^z(\ell) = \sum_{j=1}^{\ell} (-1)^j \sigma_j^z$. As shown in the introduction we have

$$P^{(u,s)}(m, t = \infty) = 2 \sum_{r \in \mathbb{Z}} P_w^{(u,s)}(r) \begin{cases} \delta(m - 2r + \ell) & \ell \text{ odd} \\ \delta(m - 2r) & \ell \text{ even} \end{cases}. \tag{69}$$

As we are dealing with an infinite temperature state, we may calculate $P_w(r)$ by using a grand canonical ensemble and working in the simultaneous eigenbasis of the $\sigma_j^z$'s. This reduces the calculation of $P_w(r)$ to the combinatorial problem of how many eigenstates there are for a given eigenvalue of $S_u^z(\ell)$ or $S_s^z(\ell)$. This is easily solved in terms of the binomial distribution

$$P_w^{(u)}(m, \infty) = P_w^{(s)}(m, \infty) = \frac{1}{2^{\ell}} \binom{\ell}{\ell/2 - m} \sim \sqrt{\frac{2}{\ell \pi}} \exp\left(-\frac{2m^2}{\ell}\right). \tag{70}$$

The result (70) for large $\ell$ is of course reproduced by applying Szegő's Lemma for block Toeplitz matrices to the determinant representations (65). This gives

$$\lim_{t \to \infty} \frac{\ln \chi^{(u)}(\lambda, \ell, t)}{\ell} = \frac{1}{2} \ln\left(\cos^2(\lambda)\right) + \mathcal{O}(1/\ell) = \lim_{t \to \infty} \frac{\ln \chi^{(s)}(\lambda, \ell, t)}{\ell}, \quad \ell \gg 1. \tag{71}$$

Fourier transforming gives the Gaussian form of $P_w^{(u,s)}(m, \infty)$ in (70).

### 5.2.2 Time dependence

The time dependence of the probability distributions for both $S_u^z(\ell)$ and $S_s^z(\ell)$ can now be determined numerically from the determinant representation (65). Results for two values of the transverse field ($h = 0.2$ and $h = 2$) are shown in Figs 14, 15, 16 and 17.

The probability distribution of $S_u^z(\ell)$ initially has a single peak at $m = 0$. At later times this peak broadens and relaxes towards the Gaussian profile (70). When quenching to the ferromagnetic phase, cf. Fig. 14, an additional feature emerges: an even/odd structure evolves at short times after the quench.

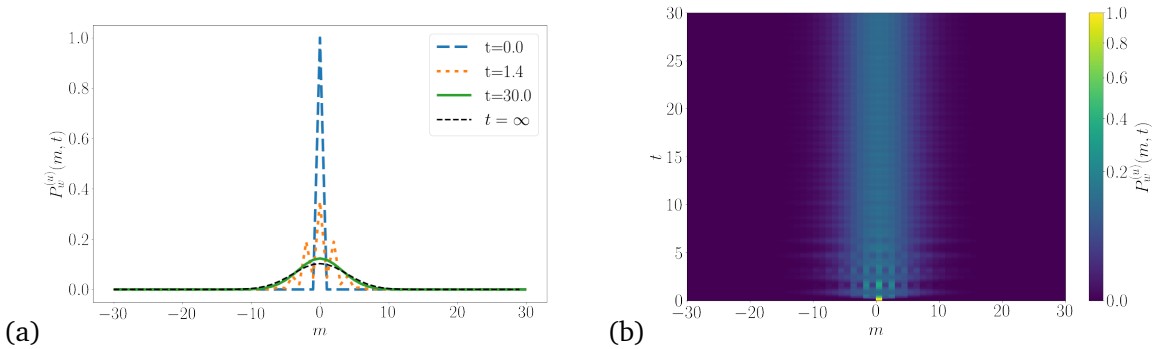

(a)                (b)

Figure 14: $P_w^{(u)}(m, t)$ for a subsystem of size $\ell = 60$ at times $t = 0, 1.4, 30.0$ for a system initialized in a Néel state and time evolved with $H(h = 0.2)$. The dotted lines are the asymptotic probability distributions given in (70).

The probability distribution of $S_s^z(\ell)$ is useful for investigating the restoration of the translational symmetry. In the initial state $P_w^{(s)}(m, t = 0)$ features a single peak at $m = -\ell/2$,

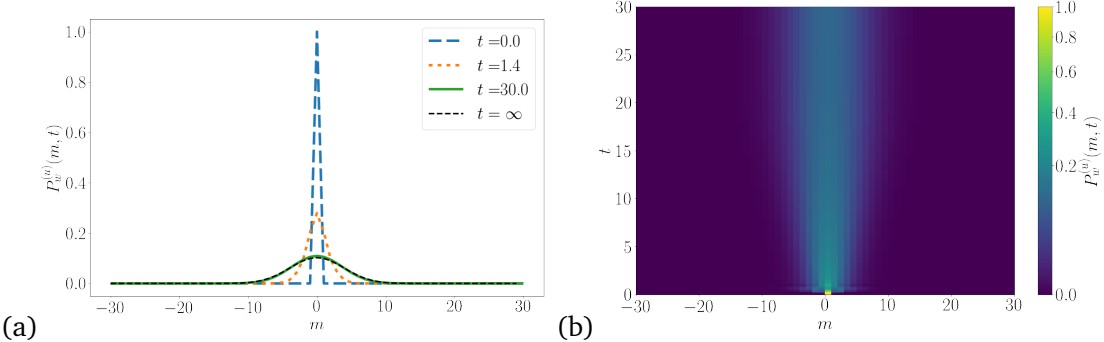

(a)                (b)

Figure 15: $P_w^{(u)}(m, t)$ for a subsystem of size $\ell = 60$ at times $t = 0, 1.4, 30.0$ for a system initialized in a Néel state and time evolved with $H(h = 2)$. The dotted lines are the asymptotic probability distributions given in (70).

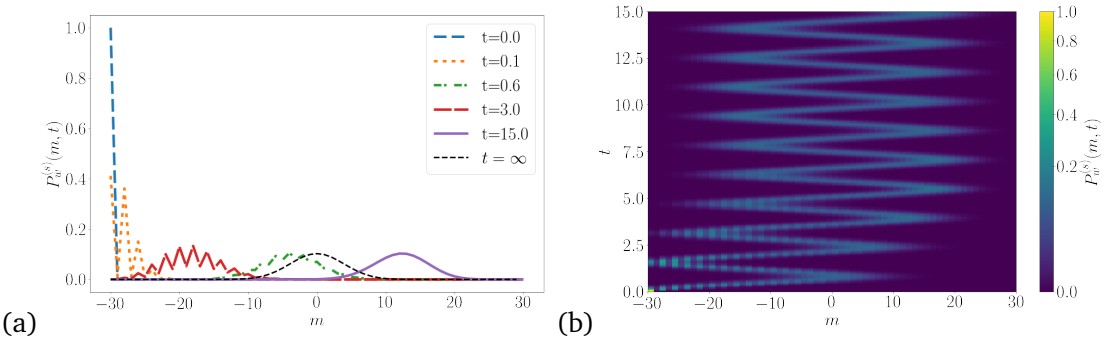

(a)                (b)

Figure 16: $P_w^{(s)}(m, t)$ for a subsystem of size $\ell = 60$ at times $t = 0, 1.4, 30.0$ for a system initialized in a Néel state and time evolved with $H(h = 0.2)$. The dotted lines are the asymptotic probability distributions given in (70).

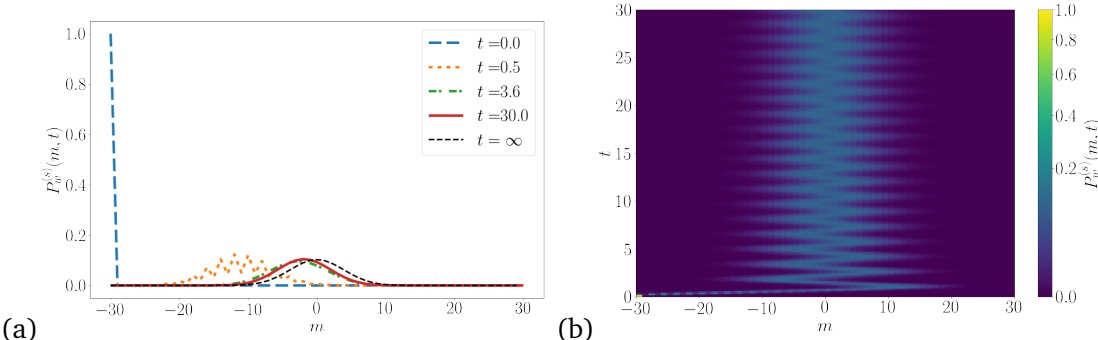

(a)                (b)

Figure 17: $P_w^{(s)}(m, t)$ for a subsystem of size $\ell = 60$ at times $t = 0, 1.4, 30.0$ for a system initialized in a Néel state and time evolved with $H(h = 2)$. The dotted lines are the asymptotic probability distributions given in (70).

which is a characteristic fingerprint of the classical Néel state (in z-direction). We first discuss quenches into the ferromagnetic phase. Here at short times after the quench $P_w^{(s)}(m, t)$ develops an even/odd structure and broadens significantly. The average of the probability distribution oscillates strongly in time and decays very slowly to its stationary value, which is a Gaussian distribution centred around $m = 0$. This shows that translational symmetry is restored very slowly.

The behaviour for quenches into the paramagnetic phase is broadly similar. An even/odd structure develops at early times, but is less pronounced that for quenches to the ferromagnetic

phase. The average of $P_w^{(s)}(m,t)$ again oscillates strongly around $m = 0$, but is seen to relax much more quickly than for quenches to the ferromagnetic phase. Approximate translational symmetry gets restored more rapidly.

# 6 Analytic results for the probability distribution

We now restrict our discussion to the particular case of transverse field quenches. As we have seen above, in this case the characteristic functions $\chi^{(u)}(\lambda, \ell, t)$ exhibit a scaling collapse at late times, *cf.* (59). This suggests that it might be possible to obtain analytic results for the late time asymptotics by a suitable generalization of the multi-dimensional stationary state approximation method previously used to determine the asymptotics of the order parameter two-point function [41] and the entanglement entropy [46]. As we will see, such a generalization is indeed possible, even though the case at hand is significantly more complicated.

Our starting point is the following expression

$$\ln \chi^{(u)}(\lambda, \ell, t) = \ell \ln(\cos \lambda) + \frac{1}{2} \text{Tr}\left(\ln(1 - \tan \lambda \, \Gamma')\right), \tag{72}$$

which is derived from (32) by using the identity $\ln(\det(A)) = \text{Tr}(\ln(A))$. The second term in (72) can be expanded in a power series

$$\frac{1}{2} \text{Tr}\left(\ln(1 - \tan \lambda \, \Gamma')\right) = -\frac{1}{2} \sum_{n=1}^{\infty} \frac{\left(\tan(\lambda)\right)^n}{n} \, \text{Tr}\left[(\Gamma')^n\right]. \tag{73}$$

This then leads us to examine integer powers $(\Gamma')^n$ of the correlation matrix. Unlike in the case of the order parameter two-point function analyzed in [42] odd powers do not vanish because $\Gamma'$ is not a real anti-symmetric matrix. The *symbol* $t'(k)$ corresponding to the correlation matrix $\Gamma'$ is defined by

$$\left(\Gamma'\right)_{ln} = \int_{-\pi}^{\pi} \frac{dk}{2\pi} e^{i(l-n)k} \, \hat{t}'(k). \tag{74}$$

Its explicit expression for a magnetic field quench from $h_0$ to $h$ is

$$\hat{t}'(k) = \begin{pmatrix} -ie^{i\theta_k}(\cos\Delta_k - i\sin\Delta_k\cos(2\varepsilon_k t)) & \sin\Delta_k\sin(2\varepsilon_k t) \\ \sin\Delta_k\sin(2\varepsilon_k t) & -ie^{-i\theta_k}(\cos\Delta_k + i\sin\Delta_k\cos(2\varepsilon_k t)) \end{pmatrix}, \tag{75}$$

where $\theta_k$ and $\Delta_k$ have been previous defined in (57). Following Ref. [42] we can represent the trace of powers of the correlation matrix as multiple integrals

$$\text{Tr}\left[(\Gamma')^n\right] = \left(\frac{\ell}{2}\right)^n \int_{-\pi}^{\pi} \frac{dk_1 \ldots dk_n}{(2\pi)^n} \int_{-1}^{1} d\xi_1 \ldots d\xi_n \, C(\vec{k}) \, F(\vec{k}) \exp\left(i\ell \sum_{j=0}^{n-1} \frac{\xi_j}{2}(k_{j+1} - k_j)\right), \tag{76}$$

where we have defined $k_0 \equiv k_n$ and

$$C(\vec{k}) = \prod_{j=0}^{n-1} \frac{k_j - k_{j-1}}{2\sin\left[(k_j - k_{j-1})/2\right]}, \qquad F(\vec{k}) = \text{Tr}\left(\prod_{i=0}^{n-1} \hat{t}'(k_i)\right). \tag{77}$$

We now change variables

$$\zeta_0 = \xi_1, \quad \zeta_i = \xi_{i+1} - \xi_i, \, i = 1, \ldots, n-1. \tag{78}$$

The integration ranges in the $\zeta$ variables is determined by the constraints

$$-1 \leq \sum_{j=0}^{k-1} \zeta_j \leq 1 , \quad k = 1, \ldots, n. \tag{79}$$

The integral over $\zeta_0$ can now be carried out as the integrand does not depend on it. This gives

$$\text{Tr}\big[(\Gamma')^n\big] = \left(\frac{\ell}{2}\right)^n \int_{-\pi}^{\pi} \frac{dk_1 \ldots dk_n}{(2\pi)^n} \int_{-1}^{1} d\zeta_1 \ldots d\zeta_{n-1} \, \mu(\vec{\zeta}) \, C(\vec{k}) \, F(\vec{k}) \exp\left(-i\ell \sum_{j=1}^{n-1} \frac{\zeta_j}{2}(k_j - k_0)\right), \tag{80}$$

where $\mu(\{\zeta\})$ is the size of the range of $\zeta_0$ under the constraints (79)

$$\mu(\vec{\zeta}) = \max\left[0, \min_{0 \leq j \leq n-1}\left(1 - \sum_{k=1}^{j} \zeta_k\right) + \min_{0 \leq j \leq n-1}\left(1 + \sum_{k=1}^{j} \zeta_k\right)\right]. \tag{81}$$

## 6.1 Multi-dimensional stationary phase approximation

For large values of $\ell$ the integrals can be carried out using a multi-dimensional stationary phase approximation. As the symbol is independent of $\zeta_j$ the stationarity conditions for the $\zeta_j$'s implies that the leading contribution to (80) derives from the region

$$k_j \approx k_0 , \quad j = 1, \ldots, n-1 . \tag{82}$$

We may thus replace $k_j$ with $k_0$ everywhere except in rapidly oscillating terms in the symbol such as $e^{2i\varepsilon(k_j)t}$. In [42] this procedure was referred to as *localization rule*. As in [41] application of this rule gives

$$C(\vec{k}) \approx 1 . \tag{83}$$

Obtaining a closed form expression for $F(\vec{k})$ is however much more involved than for the order-parameter two point function studied in Ref. [42]. We conjecture that application of the localization rule to $F(\vec{k})$ results in

$$\begin{aligned}
F(\vec{k})\Big|_{\text{loc}} &= 2 \sum_{A_1, A_2, A_3} \text{sign}(A_1 \cup A_2)(-i)^{n+S(A_1,A_2)} (\cos \Delta_{k_0})^{|A_3|}(\sin \Delta_{k_0})^{|A_1|+|A_2|} \ldots \\
\ldots &\times \cos\left([n - 2q(A_1)]\theta_{k_0} - \frac{\pi(|A_1| + |A_2|)}{2}\right) \prod_{i \in A_1} \sin(2\varepsilon(k_i)t) \prod_{j \in A_2} \cos(2\varepsilon(k_j)t)
\end{aligned} \tag{84}$$

Here the sum is over all partitions of the set of integers $\{0, 1, \ldots, n-1\}$ into three sets $A_1, A_2$ and $A_3$, where the number of elements in $A_1$ is constrained to be even. The size of the set $B = \{b_1, b_2, \ldots\}$ is denoted by $|B|$ and we have defined

$$\begin{aligned}
q(B) &= \text{mod}_n\big[\sum_{i=1}^{|B|}(-1)^{i+1} b_i\big], \\
S(A_1, A_2) &= \begin{cases} 2 & \text{if } q(A_1) \leq \frac{n}{2}, \ \text{mod}_2\big[|A_1 \cup A_2|\big] = 1 \text{ and } |A_1| > 0, \\ 0 & \text{else.} \end{cases}
\end{aligned} \tag{85}$$

Finally, $\text{sign}(A)$ is the sign of the permutation required to bring the (integer) elements of the set $A$ into ascending order. We have explicitly checked (84) for $1 \leq n \leq 15$ but have not been able to find a rigorous proof for it.

We now use the identity (for even $k$)

$$\prod_{i=1}^{k} \sin(x_i) \prod_{j=k+1}^{k+m} \cos(x_j) = \frac{(-1)^{\frac{k}{2}}}{2^{k+m}} \sum_{i_1=0}^{1} \sum_{i_2=0}^{1} \cdots \sum_{i_{k+m}=0}^{1} \exp\left(i \sum_{j=1}^{k+m} (-1)^{i_j} x_j + i\pi \sum_{j=1}^{k} i_j\right), \quad (86)$$

to rewrite the time-dependent factors in (84). This gives

$$\begin{aligned}
F(\vec{k})\Big|_{\text{loc}} = {}& 2 \sum_{A_1,A_2,A_3} \text{sign}(A_1 \cup A_2) \frac{(-i)^{S(A_1,A_2)+n+|A_1|}}{2^{|A_1|+|A_2|}} (\cos \Delta_{k_0})^{|A_3|} (\sin \Delta_{k_0})^{|A_1|+|A_2|} \cdots \\
& \cdots \times \cos\left([n-2q(A_1)]\theta_{k_0} - \frac{\pi(|A_1|+|A_2|)}{2}\right) \cdots \\
& \cdots \times \sum_{p_1=0}^{1} \cdots \sum_{p_{|A_1|+|A_2|}=0}^{1} \exp\left[2it \sum_{r=1}^{|A_1|+|A_2|} (-1)^{p_r} \varepsilon(k_{(A_1 \cup A_2)_r}) + i\pi \sum_{r=1}^{|A_1|} p_r\right], \quad (87)
\end{aligned}$$

where $(A)_r$ is the r'th element of the set $A$ and

$$(A_1 \cup A_2)_r = \begin{cases} (A_1)_r & \text{if } r \leq |A_1|, \\ (A_2)_{r-|A_1|} & \text{if } |A_1| < r \leq |A_1| + |A_2|. \end{cases} \quad (88)$$

Application of the localization rule to (80) hence results in an expression of the form

$$\begin{aligned}
\text{Tr}\left[(\Gamma')^n\right]\Big|_{\text{loc}} = {}& 2\left(\frac{\ell}{2}\right)^n \sum_{A_1,A_2,A_3} \text{sign}(A_1 \cup A_2) \cdots \\
& \cdots \times \frac{(-i)^{S(A_1,A_2)+n+|A_1|}}{2^{|A_1|+|A_2|}} (\cos \Delta_{k_0})^{|A_3|} (\sin \Delta_{k_0})^{|A_1|+|A_2|} \cdots \\
& \cdots \times \cos\left([n-2q(A_1)]\theta_{k_0} - \frac{\pi(|A_1|+|A_2|)}{2}\right) \cdots \\
& \cdots \times \sum_{p_1=0}^{1} \cdots \sum_{p_{|A_1|+|A_2|}=0}^{1} (-1)^{\sum_{r=1}^{|A_1|} p_r} \int_{-1}^{1} d\zeta_1 \ldots d\zeta_{n-1}\, \mu(\vec{\zeta}) \cdots \\
& \cdots \times \int_{-\pi}^{\pi} \frac{dk_1 \ldots dk_n}{(2\pi)^n} \exp\left[2it \sum_{r=1}^{|A_1|+|A_2|} (-1)^{p_r} \varepsilon(k_{(A_1 \cup A_2)_r}) \cdots \right. \\
& \cdots \left. -i\ell \sum_{j=1}^{n-1} \frac{\zeta_j}{2} (k_j - k_0)\right]. \quad (89)
\end{aligned}$$

In the next step we carry out a multi-dimensional stationary phase approximation for the $2n-2$ integrals over $\zeta_1, \ldots, \zeta_{n-1}$ and $k_1, \ldots, k_{n-1}$. We will assume that there is a single saddle point and use

$$\begin{aligned}
\int dx_1 \ldots dx_k\, p(x_1, \ldots, x_k) e^{i\ell q(x_1, \ldots, x_k)} \approx {}& \left(\frac{2\pi}{\ell}\right)^{k/2} \frac{p(x_1^{(0)}, \ldots, x_k^{(0)})}{\sqrt{|\det(A)|}} \cdots \\
& \cdots \times \exp\left(i\ell q(x_1^{(0)}, \ldots, x_k^{(0)}) + \frac{i\pi\sigma_A}{4}\right), \quad (90)
\end{aligned}$$

where $\sigma_A$ the signature of the matrix $A$ (i.e. the difference between the numbers of positive and negative eigenvalues), which is the Hessian of the function $q$ evaluated at the saddle point

$$A_{ij} = \frac{\partial}{\partial x_i} \frac{\partial}{\partial x_j}\bigg|_{\vec{x}=\vec{x}^{(0)}} q(x_1, \ldots, x_k). \quad (91)$$

In our case the saddle point conditions are

$$
\begin{aligned}
k_j^{(0)} &= k_0 , \quad j = 1, \ldots, n-1 , \\
\zeta_j^{(0)} &= \begin{cases} \gamma_{A_1 \cup A_2, k} & \text{if } j \in A_1 \cup A_2 , \\ 0 & \text{else} , \end{cases}
\end{aligned}
\tag{92}
$$

where $\gamma_{A,k} = \frac{4t}{\ell}(-1)^{P(A)_k^{-1}} \varepsilon'(k_0)$ and $(A_1 \cup A_2)^{-1}$ is the inverse of the index-function $(A_1 \cup A_2)_j$ defined above. The Hessian $A$ is a matrix of the form

$$
A = \frac{1}{2} \begin{pmatrix} 0 & I \\ I & M \end{pmatrix} ,
\tag{93}
$$

and hence we have $\det(A) = -4^{1-n}$ and $\sigma_A = 0$. The value of $\mu(\vec{\zeta})$ at the saddle point for a given sequence $\{p_1, p_2, \ldots, p_{|A_1|+|A_2|}\}$ is

$$
\mu(\vec{\zeta}^{(0)}) = \max\left[ 0, \min_{0 \leq j \leq |B|}\left(1 - \sum_{k=1}^{j} \gamma_{B,k}\right) + \min_{0 \leq j \leq |B|}\left(1 + \sum_{k=1}^{j} \gamma_{B,k}\right)\right] ,
\tag{94}
$$

where $B = A_1 \cup A_2 - \{0\}$. The saddle point approximation thus gives

$$
\begin{aligned}
\text{Tr}\left[(\Gamma')^n\right] &\approx \ell \sum_{A_1, A_2, A_3} \text{sign}(A_1 \cup A_2) \frac{(-i)^{S(A_1, A_2)+n+|A_1|}}{2^{|A_1|+|A_2|}} (\cos \Delta_{k_0})^{|A_3|} (\sin \Delta_{k_0})^{|A_1|+|A_2|} \ldots \\
&\ldots \times \cos\left([n - 2q(A_1)]\theta_{k_0} - \frac{\pi(|A_1|+|A_2|)}{2}\right) \sum_{p_1=0}^{1} \cdots \sum_{p_{|A_1|+|A_2|}=0}^{1} (-1)^{\sum_{r=1}^{|A_1|} p_r} \ldots \\
&\ldots \times \int_{-\pi}^{\pi} \frac{dk_0}{2\pi} \mu(\vec{\zeta}^{(0)}) \exp\left[-2it\varepsilon(k_0) \sum_{r=1}^{|A_1 \cup A_2|} (-1)^{p_r}\right].
\end{aligned}
\tag{95}
$$

The leading contribution to the final integral can then also be determined by a stationary phase approximation. This shows that all terms with $\sum_{r=1}^{|A_1 \cup A_2|}(-1)^{p_r} \neq 0$ are suppressed at late times by a factor of $1/\sqrt{t}$. Conversely, the leading contribution to $\chi^{(u)}(\lambda, \ell, t)$ at late times arises from terms with $\sum_{r=1}^{|A_1 \cup A_2|}(-1)^{p_r} = 0$, which requires $|A_1| + |A_2|$ to be even.

### 6.1.1 Structure of $\mu(\vec{\zeta}^{(0)})$

At this point it is useful to investigate the structure of $\mu(\vec{\zeta}^{(0)})$ for a given term in the multiple sum over $p_1, \ldots, p_{|A_1|+|A_2|}$ in more detail. For simplicity we focus on a particular example

$$
|A_1| = |A_2| = 2 , \quad \{p_{(A_1 \cup A_2)_k^{-1}} | k = 1, \ldots, 4\} = \{0, 1, 0, 1\}.
\tag{96}
$$

In this case we have

$$
\begin{aligned}
\mu(\vec{\zeta}^{(0)}) &= \max\left(0, \min(1, 1 - \frac{4t}{\ell}\varepsilon'(k_0)) + \min(1, 1 + \frac{4t}{\ell}\varepsilon'(k_0))\right) \\
&= \max\left(0, 2 - \frac{4t}{\ell}|\varepsilon'(k_0)|\right) = \Theta(\ell - 2|v_{k_0}|t)\left(2 - 4\frac{t|v_{k_0}|}{\ell}\right),
\end{aligned}
\tag{97}
$$

where $v_{k_0} = \varepsilon'(k_0)$ is the group velocity of Bogoliubov fermions at momentum $k_0$ and $\Theta(x)$ is the Heaviside step function. The step function in (97) is reminiscent of the light-cone structure found for two point correlation functions of local operators [66–69] and entanglement entropies [45, 70, 71]. Repeating the above exercise for

$$
|A_1| = |A_2| = m , \quad \{p_{(A_1 \cup A_2)_k^{-1}} | k = 1, \ldots, 2m\} = \{\underbrace{1, 1, \ldots, 1}_{m}, 0, 0, \ldots, 0\} ,
\tag{98}
$$

leads to the result

$$\mu(\vec{\zeta}^{(0)}) = \Theta(\ell - 2m|v_{k_0}|t)\Big(2 - 4m\frac{t|v_{k_0}|}{\ell}\Big). \tag{99}$$

All other cases can be worked out analogously and lead to Heaviside step functions $\Theta(\ell - 2m|v_{k_0}|t)$ with $m \in \mathbb{N}_0$.

## 6.2 Result for $\chi(\lambda, \ell, t)$

In order to obtain the logarithm of the characteristic function $\chi(\lambda, \ell, t)$ we now need to sum over all contributions (95) with coefficients given in (73). This is a formidable task. It turns out that the structure of Heaviside step functions discussed above provides a very useful way of organizing the complicated summation required. The full result can be expressed in the form

$$
\begin{aligned}
\ln \chi^{(u)}(\lambda, \ell, t) \quad &\approx \quad \ell \ln(\cos \lambda) + \frac{\ell}{2} \sum_{n=0}^{\infty} \int_0^{2\pi} \frac{dk_0}{2\pi} \Theta(\ell - 2n|v_k|t) \dots \\
&\dots \quad \times \Big[1 - \frac{2n|v_k|t}{\ell}\Big] \sum_{m=0}^{n+1} \cos\big(2m\varepsilon(k_0)t\big) f_{n,m}(\lambda, k_0) + \mathcal{C}.
\end{aligned}
\tag{100}
$$

Here $\mathcal{C}$ is a constant that is beyond the accuracy of the stationary phase approximation and the functions $f_{n,m}(\lambda, k_0, t)$ are given in terms of infinite series. Based on the first 15 terms in these series we conjecture the following explicit expressions

$$
\begin{aligned}
f_{0,0}(\lambda, k_0) &= 2\ln\big(1 + i\cos\Delta_{k_0}\tan\lambda e^{i\theta_{k_0}}\big), \\
f_{1,0}(\lambda, k_0) &= \ln\Bigg[1 - \frac{\sin^2\Delta_{k_0}\tan^2\lambda(\cos\theta_{k_0} + i\cos\Delta_{k_0}\tan\lambda)^2}{(\sin^2\theta_{k_0} + (\cos\theta_{k_0} + i\cos\Delta_{k_0}\tan\lambda)^2)^2}\Bigg], \\
f_{2,0}(\lambda, k_0) &= \ln\Bigg[1 + \dots
\end{aligned}
$$

$$
\dots \frac{\sin^4\Delta_{k_0}\tan^4\lambda\sin^2\theta_{k_0}(\cos\theta_{k_0} + i\cos\Delta_{k_0}\tan\lambda)^2}{((\sin^2\theta_{k_0} + (\cos\theta_{k_0} + i\cos\Delta_{k_0}\tan\lambda)^2)^2 - \sin^2\Delta_{k_0}\tan^2\lambda(\cos\theta_{k_0} + i\cos\Delta_{k_0}\tan\lambda)^2)^2}\Bigg].
\tag{101}
$$

In principle one could determine further terms $f_{n,0}$ but their contribution turns out to be negligible for all cases we have considered. The contributions $f_{n,m>0}(\lambda, k_0, t)$ are more difficult to simplify. While the term $f_{0,1}$ can still be obtained without further approximations, in order to obtain closed form expressions for $m > 1$ we have resorted to an expansion in powers of $\sin(\Delta_{k_0})$. This is expected to give very accurate results for small quenches, which are defined as producing a small density of elementary excitations through the quench [41,42]. The leading terms are then conjectured to be of the form

$$
\begin{aligned}
f_{0,1} &= -i\tan\Delta_{k_0}\ln\Bigg[\frac{1 + ie^{i\theta_{k_0}}\cos\Delta_{k_0}\tan\lambda}{1 + ie^{-i\theta_{k_0}}\cos\Delta_{k_0}\tan\lambda}\Bigg], \\
f_{1,1} &= \tan\Delta_{k_0}\Bigg(i\ln\Bigg[\frac{1 + ie^{i\theta_{k_0}}\cos\Delta_{k_0}\tan\lambda}{1 + ie^{-i\theta_{k_0}}\cos\Delta_{k_0}\tan\lambda}\Bigg] - \frac{4\cos\Delta_{k_0}\tan\lambda\sin\theta_{k_0}}{\sin^2\theta_{k_0} + \big(\cos\theta_{k_0} + i\cos\Delta_{k_0}\tan\lambda\big)^2}\Bigg) \dots \\
&\dots + \mathcal{O}(\sin^3(\Delta_{k_0})).
\end{aligned}
\tag{102}
$$

As we will see below, the contributions described by (101) and (102) are sufficient to obtain an extremely accurate description of $\chi^{(u)}(\lambda, \ell, t)$. The constant $\mathcal{C}$ can be fixed by comparing the $t \to \infty$ limit of (100) to the result obtained previously for the behaviour in the stationary state. For later convenience we define two approximations as

$$
\begin{aligned}
\ln \chi_a^{(u)}(\lambda, \ell, t) \;=\; & \ell \ln(\cos \lambda) \ldots \\
& \ldots \; + \frac{\ell}{2} \sum_{n=0}^{2} \int_0^{2\pi} \frac{dk_0}{2\pi} \Theta(\ell - 2n|v_k|t) \left[ 1 - \frac{2n|v_k|t}{\ell} \right] \ldots \\
& \ldots \; \times \sum_{m=0}^{a} \cos\left( 2m\varepsilon(k_0)t \right) f_{n,m}(\lambda, k_0) + \mathcal{C} \,,
\end{aligned}
\tag{103}
$$

where $a = 1, 2$ and where we set $f_{2,1} = 0$. The structure of the integrand in our result (100) is reminiscent of that found in connected two-point correlation functions [41] and entanglement entropies [46]. In the latter quantities it gives rise to a "light-cone" behaviour in developing connected correlations and the spreading of entanglement respectively. In contrast to these cases the expression (100) involves an infinite number of "light-cone structures" with velocities that are integer multiples of the maximum group velocity. Since the generating function involves complicated sums over multi-point correlation functions on the interval $[1, \ell]$ this is not in contradiction with the celebrated Lieb-Robinson bound [73]. The light-cone structure in connected two-point correlators and entanglement entropies can be understood in terms of simple semi-classical quasi-particle pictures [45, 66]. It would be interesting to develop an analogous understanding for the novel structure observed in the generating function, but this is beyond the scope of the present paper.

# 7 Accuracy of the asymptotic result

Our analytic result (100), (101), (102) gives the leading contributions in the *space-time scaling limit* [42] $\ell, t \to \infty$, $\ell/t$ fixed. An important question is how good this asymptotic result describes the behaviour of $\chi^{(u)}(\lambda, \ell, t)$ at small and intermediate times and subsystem sizes. In order to answer this question we now turn to a comparison between our analytical results (103) and a direct numerical evaluation of the determinant representation (32), (33). The numerical errors in the latter are negligible.

## 7.1 Small-$\lambda$ regime

A representative comparison between the analytical results $\chi_{1,2}^{(u)}(\lambda, \ell, t)$ for small values of $\lambda$ and numerics is shown in Fig 18 and 19. We see that $\chi_1^{(u)}(\lambda, \ell, t)$ reproduces the numerics very well at late times after the quench. In contrast, the oscillatory behaviour at short times is clearly not captured. The improved approximation $\chi_2^{(u)}(\lambda, \ell, t)$ (103) is seen to be in excellent agreement with the numerics.

By construction the oscillatory part of the analytic result is most accurate over the entire range of the "counting parameter" $\lambda$ when $\sin \Delta_{k_0}$ is small, i.e. for small quenches. For quenches where $\sin \Delta_{k_0}$ is no longer small we still find excellent agreement between the analytic and numerical results as long as $\tan(\lambda)$ is small. This can be understood by noting that for such values of $\lambda$ the infinite sum in (73) is dominated by the first few terms, i.e. small values of $n$. On the other hand, higher orders of $\sin \Delta_{k_0}$ only emerge for larger values of $n$. Therefore the leading order result in $\sin \Delta_{k_0}$ already provides a very good approximation in the small-$\tan(\lambda)$ regime even when $\sin \Delta_{k_0}$ is not small. This observation is of significant practical

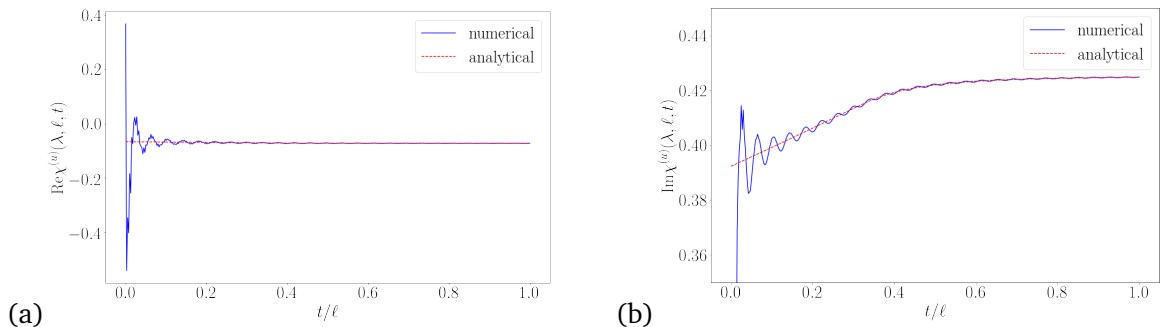

Figure 18: Real and imaginary parts of the leading approximation $\chi_1^{(u)}(\lambda = 0.1, \ell = 200, t)$ for a transverse field quench quench from $h = 0$ to $h = 0.8$. The analytic approximation gives a good description only at late times.

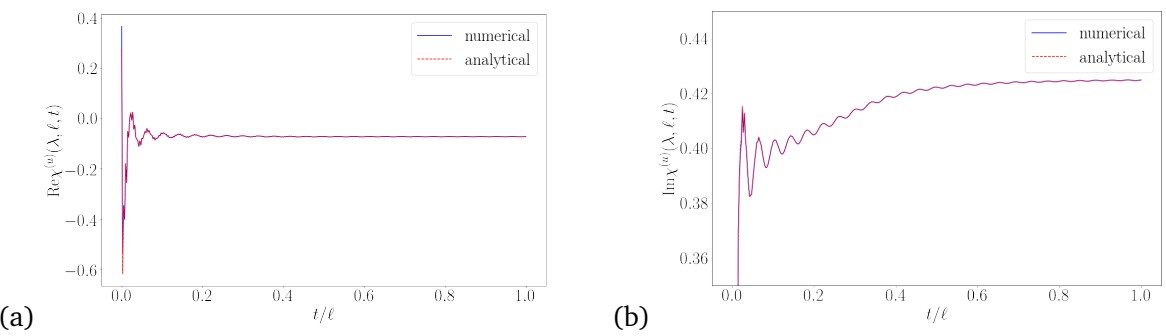

Figure 19: Real and imaginary parts of $\chi_2^{(u)}(\lambda = 0.1, \ell = 200, t)$ for a transverse field quench quench from $h = 0$ to $h = 0.8$. The analytical expression (red dashed line) is seen to be in excellent agreement with the numerical results, which have negligible errors on the scale of the figure.

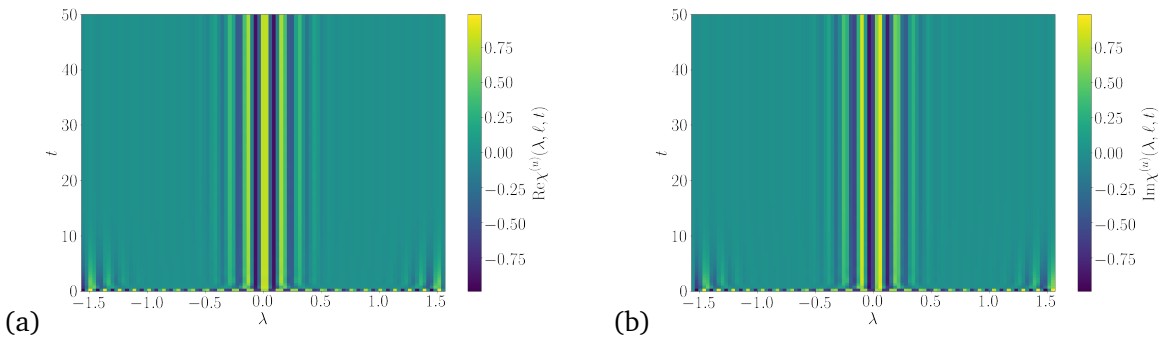

Figure 20: (a) Real and (b) imaginary parts of $\chi^{(u)}(\lambda, \ell = 50, t)$ as functions of $\lambda$ and $t$ for a transverse field quench from $h = 5$ to $h = 1.5$. We observe that the characteristic function is small unless $\lambda$ is small. The behaviour for quenches within the ferromagnetic phase and quenches between the phases is similar.

importance: As shown in Fig. 20 in a particular example $|\text{Re}\chi^{(u)}(\lambda, \ell, t)|$ and $|\text{Im}\chi^{(u)}(\lambda, \ell, t)|$ are largest in the vicinity of $\lambda = 0$ (except at short times). This implies that the corresponding probability distribution, which is the object we are ultimately interested in, will be dominated by the small-$\lambda$ regime. As a consequence (100), (101), (102) provide a good approximation for the calculation of $P_w^{(u)}(m)$ for all quenches.

## 7.2 Large-$\lambda$ regime

In the large-$\lambda$ regime we have to distinguish between the cases where the symbol in the stationary state has zero or non-zero winding number, *c.f.* section 5.1.1. The first case covers quenches to the paramagnetic phase. Here we find that our analytic result is again in good agreement with numerics. The second scenario applies to quenches to the ferromagnetic phase and $\lambda > \lambda_c(h_0, h)$. We have shown in section 5.1.2 that there is no good scaling collapse in this regime of counting parameters for the moderate subsystem sizes and times of interest here. It should therefore not come as a surprise that the asymptotic result does not provide a good approximation in this regime. Presumably (100), (101), (102) no longer hold in this regime because the analytic continuation of the power series expansion of the logarithm (73) becomes non-trivial in this case. In practice the failure of the analytic approach to give a good account of the generating function in this parameter regime is irrelevant as $\chi^{(u)}(\lambda, \ell, t)$ itself is extremely small and makes a negligible contribution to the probability distribution. As shown in Fig. 21, the main contribution to the latter, which after all is our object of interest, arises from the small-$\lambda$ regime of the generating function, which is well approximated by our analytic expressions.

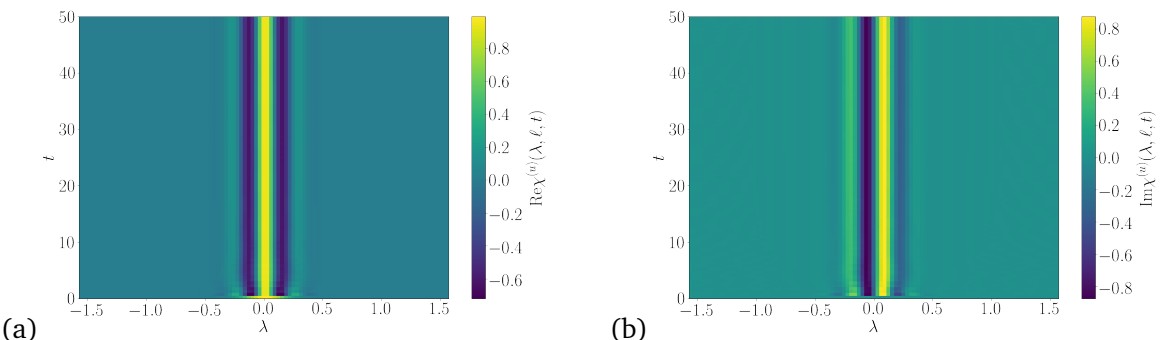

Figure 21: Real (a) and imaginary (b) parts of $\chi^{(u)}(\lambda, \ell = 50, t)$ for a quench within the ferromagnetic phase from $h_0 = 0.2$ to $h = 0.8$. The dominant contribution to the probability distribution arises from the small-$\lambda$ regime.

## 7.3 Relative errors

In order to provide a more quantitative discussion of the quality of the approximate results (103) we consider the relative errors

$$r_{1,2}(\lambda, \ell, t) = \left| 1 - \frac{\ln\left(\chi^{(u)}_{1,2}(\lambda, \ell, t)\right)}{\ln\left(\chi^{(u)}_{\text{num}}(\lambda, \ell, t)\right)} \right| , \tag{104}$$

where $\chi^{(u)}_{1,2/\text{num}}(\lambda, \ell, t)$ are respectively the analytic approximations (103) and the result of the numerical computation of the determinant representation (32), (33). In Fig. 22 we plot the time dependence of the relative errors for a quench from $h_0 = 5$ to $h = 1.5$ for a subsystem of size $\ell = 200$ and two values of the counting parameter $\lambda$. The maximal value of $\sin(\Delta_{k_0})$ within the domain of integration approximately 0.54, which means that higher orders in $f_{1,1}$ can be important. As we have argued above, this will be the case if $\tan(\lambda)$ is not small. In Fig. 22 (a) $\lambda = 0.1$ is taken to be small, and the quality of both approximations $\chi^{(u)}_{1,2}(\lambda, \ell, t)$ is seen to be excellent. In Fig. 22 (b) the counting parameter $\lambda = 1.4$ is taken to be large. This leads to a significantly larger error, which is however still fairly small and also decays in time. We see that the analytic results provide a good approximation for all values of $\lambda$.

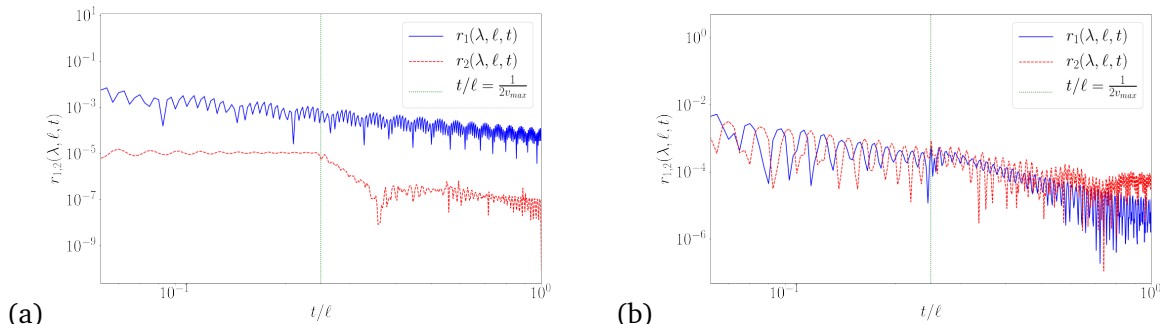

Figure 22: (a) Relative errors $r_{1,2}(\lambda = 0.1, \ell = 200, t)$ for a quench within the paramagnetic phase from $h_0 = 5$ to $h = 1.5$. (b) same for $r_{1,2}(\lambda = 1.4, \ell = 200, t)$.

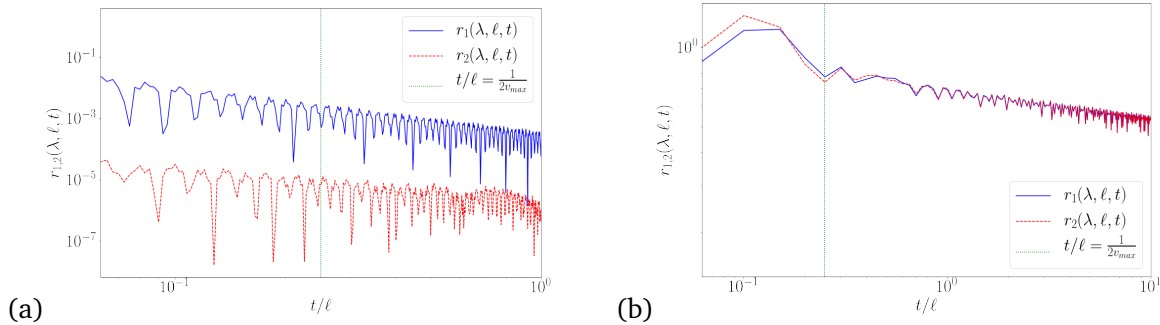

Figure 23: (a) Relative errors $r_{1,2}(\lambda = 0.1, \ell = 200, t)$ for a quench within the ferromagnetic phase from $h_0 = 0$ to $h = 0.8$. (b) same for $r_{1,2}(\lambda = 1.4, \ell = 200, t)$.

We now turn to a parameter regime, in which our analytic results no longer provide a uniformly good approximation for all values of the counting parameter $\lambda$. Fig. 23 shows results for a quench from $h_0 = 0.2$ to $h = 0.8$. The maximal value of $\sin\Delta_{k_0}$ in the integration range is now 0.71 so that higher orders in $f_{1,1}$ can again be important. For small values of $\lambda$ the relative errors of both analytical approximations are small and decreasing in time. On the other hand $\chi_{1,2}^{(u)}(\lambda, \ell, t)$ cease to provide accurate approximations for large values of $\lambda$ with $\lambda > \lambda_c(h_0, h)$ as can be seen in Fig. 23 (b). However, we want to stress once more that $\chi^{(u)}(\lambda, 200, t)$ itself is extremely small in this parameter regime and makes only a negligible contribution to the probability distribution.

## 7.4 Probability distributions

An asymptotic expansion for the probability distribution $P_w^{(u)}(m, t)$ can be obtained by Fourier transforming the generating function, *cf.* Eq. (14). As expected on the basis of the discussion above, we find that the analytic result becomes very accurate at sufficiently late times for *all* quenches. At intermediate and short times we still find excellent agreement between the analytical and numerical results for quenches originating the ferromagnetic, see e.g. Fig. 24 (a). For quenches from the paramagnetic phase the analytic result is an excellent agreement with numerics at short and intermediate times as long as the quench is "small". In practice this covers all quenches within the paramagnetic phase as long as $h$ is not very close to 1. For other quenches the corrections to the $f_{1,1}$ term in (102) will become significant at short and intermediate times.

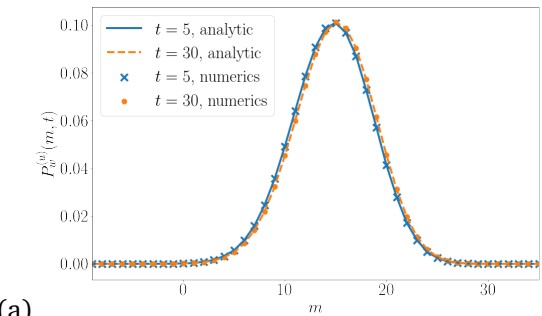
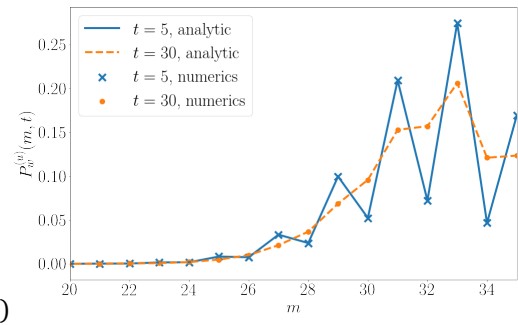

Figure 24: Comparison of the asymptotic expression for $P_w^{(u)}(m,t)$ obtained from eqns (100), (101), (102) (solid lines) to numerics (symbols) for transverse field quenches with (a) $h_0 = 0.2$ and $h = 0.8$ and (b) $h_0 = 5$ and $h = 2$. The agreement is seen to be excellent.

# 8 Conclusions

We have analyzed the full counting statistics of the transverse and staggered magnetization of a subsystem in the thermodynamic limit of the transverse field Ising chain. We derived a convenient determinant representation for the corresponding generating functions $\chi^{(u,s)}(\lambda, \ell, t)$. We first considered the FCS in equilibrium states and showed that the probability distributions are always non-Gaussian except in the limit of infinite subsystem size at finite temperature. We determined the temperature and field dependence of the generating function as well as the first few cumulants. We then moved on to the main focus of our work, the calculation of the FCS after quantum quenches. We considered two quench protocols: transverse field quenches and evolution starting from a classical Néel state. We first determined the FCS in the stationary states reached at late times. The probability distributions are again non Gaussian, except in the limit of infinite subsystem size. We analyzed the time evolution of the probability distributions $P^{(u,s)}(m,t)$ for a variety of quenches by numerically evaluating the exact determinant representation for the generating function (the numerical errors incurred are negligible). For transverse field quenches originating in the paramagnetic phase $P^{(u,s)}(m,t)$ showed interesting smoothing and broadening behaviour in time. In contrast, $P^{(u,s)}(m,t)$ displayed a simpler behaviour for quenches originating in the ferromagnetic phase. In the case of a Néel quench $P^{(s)}(m,t)$ encoded detailed information on the restoration of translational invariance. The numerical approach provided us with evidence for the existence of a *scaling regime* for the generating function in which we observed data collapse according to the scaling form (59). This is turn allowed us to proceed with the derivation of the main result of our work: the analytic expression (100) for the FCS after transverse field quenches in the *space-time scaling limit* $t, \ell \to \infty$, $t/\ell$ fixed. This was achieved by a substantial generalization of the multi-dimensional stationary phase approximation method of Refs [42, 46]. We performed a careful comparison of our analytic results to numerics (that has negligible errors) and found excellent agreement on the level of the probability distributions for all cases considered. We observed that the expression for the generating function exhibits an interesting multiple light-cone structure that has no analog in either correlation functions of local observables [41] or in the entanglement entropy [46]. An interesting open question is whether this structure can be understood in terms of the kind of semiclassical quasi-particle picture that has been successfully employed to explain the main features observed in the dynamics of both entanglement [45] and correlations [66].

Our work provides the first analytic results for FCS after quantum quenches and hopefully will pave the way for further studies. Here we have focussed on the FCS for the transverse

magnetisation. It would be very interesting to determine the FCS for the longitudinal magnetisation, which is the order parameter characterising the Ising quantum phase transition. A more straightforward but interesting extension would be to study certain observables in free fermion models with long-range hopping and/or pairing [72, 74, 75]. Similarly, the probability distribution of the (smooth) subsystem magnetisation in the spin-1/2 Heisenberg XXZ chain should be calculable both at finite temperatures [76] and in the stationary states after certain quantum quenches [77–85]. For quantum quenches in the regime where bosonization provides a good approximation [86] the full time evolution of the probability distribution for certain observables can be obtained in a straightforward way. Finally, the case of integrable chains of higher spin could be studied both in equilibrium [87,88] and after a quench [84,89].

## 9 Acknowledgements

This work was supported by the EPSRC under grant EP/N01930X (FHLE) and by the Clarendon Scholarship fund (SG). We are grateful to the Erwin Schrödinger International Institute for Mathematics and Physics for hospitality and support during the programme on *Quantum Paths*.

## A  Asymptotics of block Toeplitz matrices

Let $T_\ell$ be a general block Toeplitz matrix with elements $(T_\ell)_{ln} = t_{l-n}$. The *symbol* $\tau(e^{ik})$ of $T_\ell$ is defined by

$$t_n \equiv \int_0^{2\pi} \frac{dk}{2\pi} \tau(e^{ik}) e^{-ink}. \tag{105}$$

In cases where the symbol has winding number zero, the large-$\ell$ asymptotics of the determinant of $T_\ell$ is (under certain conditions) given by [90]

$$\ln \det(T_\ell) = \ell \int_0^{2\pi} \frac{dk}{2\pi} \ln \det\left(\tau(e^{ik})\right) + \det\left(T(\tau^{-1})T(\tau)\right) + o(1). \tag{106}$$

Here $T(\tau)$ denotes an infinite Toeplitz matrix with symbol $\tau$. In the case where the block-size is 1, this reduces to the Szegő limit theorem

$$\ln \det(T_\ell) = \ell \int_0^{2\pi} \frac{dk}{2\pi} \ln \tau(e^{ik}) + \sum_{q \geq 1} q \, (\ln \tau)_q (\ln \tau)_{-q} + o(1), \tag{107}$$

where

$$(\ln \tau)_q = \int_0^{2\pi} \frac{dk}{2\pi} \ln \tau(e^{ik}) e^{-ikq} . \tag{108}$$

The large $\ell$ asymptotics of Toeplitz determinants in cases where the symbol $\tau$ has winding number $\pm 1$ is given by [42,91]

$$\ln \det(T_\ell) = \ell \int_0^{2\pi} \frac{dk}{2\pi} \ln a(e^{ik}) + \sum_{q \geq 1} q \, (\ln a)_q (\ln a)_{-q} + \ln \int_0^{2\pi} \frac{dk}{2\pi} e^{-i\ell k} \frac{a_-(e^{ik})}{a_+(e^{ik})} + o(1) , \tag{109}$$

where

$$a(e^{ik}) \equiv -e^{\mp ik} \tau(e^{ik}) = \exp\left(\sum_{j=1}^\infty (\ln a)_{\pm j} e^{\pm ijk}\right) . \tag{110}$$

# B  Perturbation theory around the $h \to \infty$ limit

We have seen that the probability distributions $P^{(u,s)}(m, t)$ exhibit an even/odd structure in $m$ for short times after quenches starting in the paramagnetic phase. In this appendix we show that this structure can be understood in perturbation theory around the $h \to \infty$ limit. For simplicity we consider the probability distribution $P^{(u)}(m)$ in the ground state at $h \gg 1$. In the limit $h \to \infty$ the ground state is the saturated ferromagnetic state along the transverse field direction

$$|0\rangle^{(0)} = |\uparrow \ldots \uparrow\rangle . \tag{111}$$

Hence

$${}^{(0)}\langle 0|e^{i\lambda S_u^z(\ell)}|0\rangle^{(0)} = e^{i\lambda \ell} . \tag{112}$$

The corresponding probability distribution is a delta function at $m = -\ell/2$. The other eigenstates of $\sum_j \sigma_j^z$ are denoted by $|n\rangle^{(0)}$. The leading correction to the generating function arises at second order in perturbation theory in $H_1 = \sum_j \sigma_j^x \sigma_{j+1}^x$. The relevant corrections to the ground state are

$$|0\rangle^{(2)} = |0\rangle^{(0)} + \sum_{n \neq 0} |n\rangle^{(0)} \frac{{}^{(0)}\langle n|H_1|0\rangle^{(0)}}{E_0^{(0)} - E_n^{(0)}} - \frac{1}{2}|0\rangle^{(0)} \sum_{n \neq 0} \frac{\left|{}^{(0)}\langle n|H_1|0\rangle^{(0)}\right|^2}{(E_n^{(0)} - E_0^{(0)})^2} + \ldots \tag{113}$$

Substituting this into the expression for the generating function gives

$$
\begin{aligned}
{}^{(2)}\langle 0|e^{i\lambda S_u^z(\ell)}|0\rangle^{(2)} &= e^{i\lambda \ell} \left[ 1 - \sum_{n \neq 0} \frac{\left|{}^{(0)}\langle n|H_1|0\rangle^{(0)}\right|^2}{(E_n^{(0)} - E_0^{(0)})^2} \right] \ldots \\
&\ldots \; + \sum_{n \neq 0} {}^{(0)}\langle n|e^{i\lambda S_u^z(\ell)}|n\rangle^{(0)} \left| \frac{{}^{(0)}\langle n|H_1|0\rangle^{(0)}}{E_0^{(0)} - E_n^{(0)}} \right|^2 .
\end{aligned}
\tag{114}
$$

In order for ${}^{(0)}\langle n|H_1|0\rangle^{(0)}$ to be non-zero the product state $|n\rangle^{(0)}$ must have precisely two overturned spins. Let us denote their positions by $j$ and $j + 1$. For $\ell \geq 2$ we then have

$$
{}^{(0)}\langle n|e^{i\lambda S_u^z(\ell)}|n\rangle^{(0)} = \begin{cases} e^{i\lambda(\ell - 4)} & \text{if } 1 \leq j < \ell \\ e^{i\lambda(\ell - 2)} & \text{if } j = 0 \text{ or } \ell \\ e^{i\lambda \ell} & \text{else.} \end{cases}
\tag{115}
$$

This gives

$$
{}^{(2)}\langle 0|e^{i\lambda S_u^z(\ell)}|0\rangle^{(2)} = e^{i\lambda \ell} \left[ 1 - \frac{\ell + 1}{16h^2} \right] + \frac{2}{16h^2} e^{i\lambda(\ell - 2)} + \frac{\ell - 1}{16h^2} e^{i\lambda(\ell - 4)} .
\tag{116}
$$

The corresponding probability distribution is

$$
P^{(u)}(m)\Big|_{\text{PT}} = \left[ 1 - \frac{\ell + 1}{16h^2} \right] \delta(m - \ell/2) + \frac{2}{16h^2} \delta(m + 1 - \ell/2) + \frac{\ell - 1}{16h^2} \delta(m + 2 - \ell/2).
\tag{117}
$$

This is seen to exhibit an even/odd effect as the corrections for $m = \ell/2 \bmod 2$ are proportional to the subsystem size.

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
