# Peer review of "Full Counting Statistics in the Transverse Field Ising Chain"

_SciPost Physics, doi:SciPost Phys. 4, 043 (2018)_

## Round 2 · Referee Report · Anonymous · 2018-5-16

Strengths

1. Technically strong
2. Well-written and motivated
3. First work to tackle analytically full counting statistics in an out-of-equilibrium situation
4. Resuls tested against extensive numerics

Weaknesses

1. The presentation could be made clearer (see Requested changes)
2. Some results would deserve further discussion

Report

This work studies the full counting statistics (FCS) of sub-regions magnetizations in the transverse field Ising chain, both at thermal equilibrium and following a quantum quench.
While much attention has been brought over the recent years to the expectation values of local observables, the full-counting statistics is of major importance both theoretically and experimentally.
An important fact used throughout the paper is that the generating functions for full-counting statistics of the observables considered here can be brought to a gaussian form in the free-fermionic formulation, and can therefore be tackled analytically. In particular, the authors present results for the FCS at thermal equilibrium, as well as an analytical description of the large time, large scale behaviour of the FCS after a quantum quench. Both these results are new, and open the gate for further developments.
I recommend publication in SciPost, once the following issues have been addressed :

Requested changes

1. The definitions of some relevant object are scattered in different parts of the paper, which makes the reading a bit uneasy. In particular, the notion of symbol for Toeplitz matrices and the associated winding number is used in eq. (47), but defined only in Appendix A, which is refered to afterwards. I would suggest to define those in a clearer fashion, and for generic Toeplitz matrices, in the bulk of the paper.

2. In section III, the authors introduce a representation of the lattice spin operators in terms of a set of Majorana fermions. It would be useful to have the relation between those and the Jordan-Wigner fermions of section II.A clarified.

3. The multiple light-cone structure exhibited in section VI deserves to be commented further. Is there an interpretation of the associated velocities in terms of quasiparticles, for instance ?

4. Some minor typos and comments :
- section III, after eq. (16): double occurrence of "the" in the sentence "Hence they are univocally determined by the the correlation matrices...."
- section V.B : "We now turn to the time evolution of $P_w^{(u)}(m,t)$" : $P_w^{(s)}(m,t)$ should be mentioned as well, since it is studied in this section
-conclusion : missing word in the sentence "A more straightforward but interesting extension would be to certain observables...."

---

## Round 2 · Referee Report · Anonymous · 2018-5-17

Strengths

(1) detailed calculations.
(2) existence of a scaling collapse.
(3) detailed calculations.

Weaknesses

(1) No independent numerical verification.
(2) Misses previous work on non-equilibrium FCS of charge transport in the IRLM.

Report

The authors report on the full counting statistics (FCS) in the transverse field Ising chain.
The manuscript appears to be interesting and impressing and I recommend the work for publication,
provided the authors take the comments into account.

In order to judge the results one has to rely on the analytic calculations.
While the manuscript is an analytic work, an -- from the analytical work --
independent numerical check would be helpful to address a larger audience.

For example in work on the FCS of charge transport within the interacting resonant level model (IRLM),
see PRL 107, 206801 (2011), PRB 89, 081401 (2014), Phys. Scr. 2015, 014009 (2015), the analytic
results are accompanied by independent numerics. These tests make the results plausible for
reader not familiar with the analytical methods. To the best of my understanding
the numerical tests presented in this work, like in Fig. 22, only test the validity of approximations,
not of the complete approach. I understand that this would represent a lot additional work, therefore I'm
only suggesting it. However, the authors should at least point out, that this kind of verification
is possible. Specifically, this could be done by a time-dependent Bogoliubov - de Gennes approach
without having to rely on methods for strongly correlated systems.
In addition, the cited papers provide an earlier work non-equilibrium FCS from quantum quenches.

Finally, the authors may want to acknowledge PRL 107, 100601 (2011).

Remarks:
-> Eq. (2) misses parentheses in last term
-> page 2: "the the " --> "the"

Requested changes

(1) Address the work on FCS of charge transport. It would be nice if the authors could compare the finite time corrections in the IRLM with the findings in the current manuscript.

(2) Possibly providing an independent numerical check.

(3) Correcting the typos.

---

## Round 2 · Referee Report · Anonymous · 2018-5-31

Strengths

1. Clear presentation
2. Technically strong
3. Particularly useful for future studies of non-equilibrium properties of quantum spin systems

Weaknesses

1. A short discussion of the theoretical and experimental implications of the main results could be helpful

Report

In this manuscript, the authors investigate the full counting statistics of the transverse magnetization of a subsystem. A determinant representation of the generating functions is derived both for equilibrium states and for the time evolution following quantum quenches starting from different initial states. The asymptotic analytical results are carefully compared to exact numerical results and an excellent agreement is found.

The manuscript contains important and novel analytical results that are particularly useful for future studies of non-equilibrium properties of quantum spin systems. Moreover the presentation is very clear and the derivations are given is sufficient detail.

I strongly recommend the publication of this manuscript in SciPost.

Requested changes

1. Optional change: Given the length of the manuscript, a short summary of the main results might be helpful for some readers (although this is a matter of taste).

---

## Round 3 · Author Response

Referee 1:

We thank the referee for their careful reading of our manuscript and for the constructive comments. In the following we reply to the questions s/he raised.

  1. "The definitions of some relevant object are scattered in different parts of the paper, which makes the reading a bit uneasy. In particular, the notion of symbol for Toeplitz matrices and the associated winding number is used in eq. (47), but defined only in Appendix A, which is refered to afterwards. I would suggest to define those in a clearer fashion, and for generic Toeplitz matrices, in the bulk of the paper."

We thank the referee for pointing out the missing definition of the symbol of Toepliz matrices. We added this definition as Eq. (47).

  1. "In section III, the authors introduce a representation of the lattice spin operators in terms of a set of Majorana fermions. It would be useful to have the relation between those and the Jordan-Wigner fermions of section II.A clarified."

We thank the referee for pointing out the missing definition of the relation between the complex fermions and Majorana fermions. We added it in the text after Eq. (18).

  1. "The multiple light-cone structure exhibited in section VI deserves to be commented further. Is there an interpretation of the associated velocities in terms of quasiparticles, for instance ?"

We agree with the referee that a comment on the light-cone structure in the discussion of the result is in place and have added a paragraph at the end of chapter VI.B. We agree that finding a quasi-particle interpretation would be interesting, but as this does not seem a trivial task we leave it for future work.

  1. "Some minor typos and comments :
  2. section III, after eq. (16): double occurrence of "the" in the sentence "Hence they are univocally determined by the the correlation matrices...."
  3. section V.B : "We now turn to the time evolution of P_w^(u)(m,t): P_w^(s)(m,t) should be mentioned as well, since it is studied in this section -conclusion : missing word in the sentence "A more straightforward but interesting extension would be to certain observables...." "

We thank the referee for pointing out the typos and further comments. We have incorporated the changes in the current version.

%%%%%%%%%%%%%%%%%%%%%%%%%%%%%%%%%%%%%%%%%%%%%%%%%%%%%%%%%%%%%%%%%%%%%%%%%%%%%

Referee 2:

The referee requested the following changes:

(1) "Address the work on FCS of charge transport. It would be nice if the authors could compare the finite time corrections in the IRLM with the findings in the current manuscript."

While we share the referee's opinion that the FCS of charge transport in quantum impurity problems is very interesting, we fail to see its relevance to the problem considered in our work. We determine the FCS of the transverse magnetization in a large, finite subsystem after a global quantum quench in a homogeneous system, which in our understanding is an altogether different problem. We therefore see no reason to either discuss the IRLM or attempt to provide a list of references for the analysis of FCS in such kinds of problems.

(2) "Possibly providing an independent numerical check."

We are extremely puzzled by this comment. As stated in the paper we have conducted extensive numerical tests of the analytic results, and some representative comparisons are in fact shown in the manuscript. For example Fig. 22 shows a comparison between the analytic calculation, which is based on an asymptotic expansion, to numerically exact results (no approximations) in the thermodynamic limit (L=\infty).

(3) "Correcting the typos."

We are grateful to the referee for pointing out these typos, which we have corrected.

%%%%%%%%%%%%%%%%%%%%%%%%%%%%%%%%%%%%%%%%%%%%%%%%%%%%%%%%%%%%%%%%%%%%%%%%%%%%%

Referee 3:

We thank the referee for his/her helpful comments.

  1. "Optional change: Given the length of the manuscript, a short summary of the main results might be helpful for some readers (although this is a matter of taste)."

We have extended the summary of the paper along the lines suggested by the referee.

---

## Round 3 · List of Changes

- Added definition of symbol of Toeplitz matrices (Eq. 47)
- Added relation between complex fermions and Majoranas
- Added paragraph with comment on multiple lightcone structure at end of chapter VI.B
- Extended summary
- Fixed typos

You are currently on this page

Resubmission 1803.09755v3 on 13 June 2018

---

## Editorial Decision

published